# RLZero: Direct Policy Inference from Language Without In-Domain Supervision

**Harshit Sikchi**[*,1], **Siddhant Agarwal**[*,1], **Pranaya Jajoo**[*,2], **Samyak Parajuli**[*,1], **Caleb Chuck**[*,1],
**Max Rudolph**[*,1], **Peter Stone**[†,1,3], **Amy Zhang**[†,1], **Scott Niekum**[†,4]
[1] The University of Texas at Austin, [2] University of Alberta
[3] Sony AI, [4] UMass Amherst

## Abstract

The reward hypothesis states that all goals and purposes can be understood as the maximization of a received scalar reward signal. However, in practice, defining such a reward signal is notoriously difficult, as humans are often unable to predict the optimal behavior corresponding to a reward function. Natural language offers an intuitive alternative for instructing reinforcement learning (RL) agents, yet previous language-conditioned approaches either require costly supervision or test-time training given a language instruction. In this work, we present a new approach that uses a pretrained RL agent trained using only unlabeled, offline interactions—without task-specific supervision or labeled trajectories—to get zero-shot test-time policy inference from arbitrary natural language instructions. We introduce a framework comprising three steps: *imagine*, *project*, and *imitate*. First, the agent imagines a sequence of observations corresponding to the provided language description using video generative models. Next, these imagined observations are projected into the target environment domain. Finally, an agent pretrained in the target environment with unsupervised RL instantly imitates the projected observation sequence through a closed-form solution. To the best of our knowledge, our method, `RLZero`, is the first approach to show direct language-to-behavior generation abilities on a variety of tasks and environments without any in-domain supervision. We further show that components of `RLZero` can be used to generate policies zero-shot from cross-embodied videos, such as those available on YouTube, even for complex embodiments like humanoids.

## 1 Introduction

Underlying the many successes of RL lies the engineering challenge of reward design. It often takes a skilled expert to engineer a reward function which still might not be fully aligned to the true task specification [9]. Not only does this challenge restrict the scaling of RL agents, but it also makes those agents inaccessible to any user inexperienced in reward design. Even for experts, confidently specifying reward functions is generally infeasible because learning agents have often been found to hack them [38, 4, 18, 31]; i.e. the learned policies for the reward function produce behaviors that do not align with what the human intended. Language is an expressive communication channel for human intent and allows bypassing traditional reward design, but learning a mapping from language to behaviors has historically required collecting and annotating behaviors that correspond to language [26, 33, 54] in the domain where the agent is to be deployed. Scaling this strategy is impractical as it requires labeling the agent's large space of behaviors with corresponding language descriptions, and repeating the process in every new domain.

This paper is motivated by the vision that a true generalist agent should be able to solve new tasks that are presented to it in a way that is natural for people to specify. Language offers an intuitive channel

---

* Equal contribution,[†] Equal Advising. Correspondence to hsikchi@utexas.edu

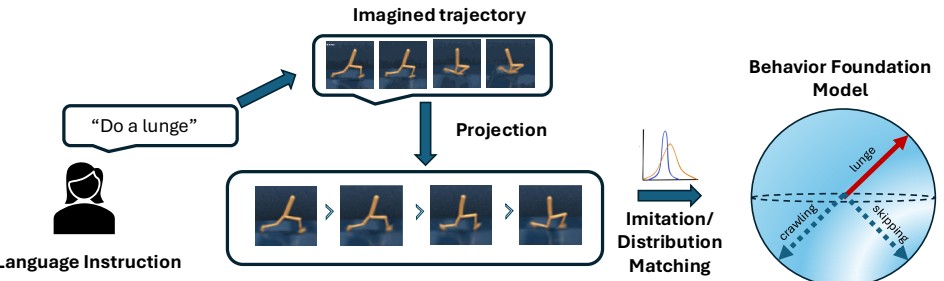

Figure 1: `RLZero` framework of **imagine**, **project**, and **imitate**: A video trajectory is imagined using the text prompt and each frame is projected to agent's observation space. A closed form solution to imitation learning for BFMs trained with unsupervised RL is used to obtain a policy that mimics the projected video behavior.

for task specification, but prior language-based policy learning methods have required significant agent training for specific language commands. In contrast, the objective of this work is to enable generalist agents to translate language commands into behaviors, with no post-command training. The solution we identify can be divided into two parts: (1) being able to communicate the task specification through language commands, and, (2) quickly producing policies for the inferred task. Large-scale multimodal foundation models [79] provide us with the first part of the solution. Trained on large amounts of internet data, they can generate video segments that communicate what performing a task entails. Unfortunately, these video segments can be out-of-distribution of the agent's domain with different dynamics and embodiments. Thus, this work addresses this challenge by projecting individual frames of generated video to agent observations using semantic similarity [61, 89], thus grounding them in the agent's observation space.

This presents us with the next part of the problem: How do we quickly infer a policy that solves the communicated task i.e. mimics the grounded imagined trajectories? We utilize the recent success of successor-features-based unsupervised RL methods, sometimes called Behavior Foundation Models (BFMs) [76, 55, 1, 67] for direct policy inference. These models are pretrained in the agent's domain using task-agnostic prior interactions of *arbitrary quality* from the environment to encode a wide variety of behaviors. During inference, near optimal policies corresponding to *any reward function* can be obtained in a zero-shot manner, i.e. in closed form without any gradient updates.

Putting these two parts together, we frame the problem of language-to-policy as a translation from one space, of language, to another space, of behaviors, as specified by successor features. To address this problem, we leverage the capabilities both of video language models to act as a pretrained medium for translating language to states, and of unsupervised RL methods to provide a zero-shot solution for distribution matching. The resulting approach is inspired by the imagination capabilities of humans to picture in their mind possibilities in the real world [63, 64, 59] and then ro rely on past experiences, memories, and abilities to inform their actions. Our framework `RLZero` (illustrated in figure 1) comprises three main steps: a) **Imagine:** Imagine trajectories given a language command; b) **Project**: The frames of imagined trajectories are projected to real observations of the agent; c) **Imitate:** Leverage a pretrained BFM to directly output a policy that matches the state visitation distribution of the imagined trajectories. Our experiments indicate that RLZero is a promising approach to designing an interpretable link connecting humans to RL agents. We find that RLZero is an effective method on a variety of tasks where reward function design would require an expert reward engineer. We further find that RLZero exhibits successful direct cross-embodiment transfer, the first approach to be able to do so, to the best of our knowledge. This paper's main contribution is the framework of imagine, project, and zero-shot imitate, which diverges from the prior approach of using VLMs as reward functions—which can be hacked—and instead focuses on zero-shot imitation with unsupervised RL, which admits a unique solution that matches the imagined behavior.

## 2    Related Work

**Language and Vision Based Control:** There has been a rich history of using vision [13, 32, 68, 45] and language [73, 69, 27, 48] as task primitives in RL. While several of these infer a reward function from language instructions and/or VLMs [62, 6, 68, 85, 88, 43] and require training an RL agent using this reward, others have used trajectory-instruction pairs to train the agents with the hope

of generalizing to related instructions. A number of recent works have utilized large pretrained foundation models to generate video plans [16, 17, 44] or plans containing textual actions [29, 3, 80]. Some similar methods generate these video plans using diffusion models or energy models [43]. But these methods often require in-domain labeled trajectories to finetune the large models or train these diffusion models [43, 44]. These often rely on APIs [39], inverse dynamics models [16, 44], or goal-conditioned policies trained using text-expert trajectory pairs [17] to convert these plans to actions in the agent's space. This means that these methods obtain an open-loop sequence of actions that is a translation from the video or textual plan to the agent's action space. RLZero differs from all of these as: (1) It does not have any test time training or planning and directly obtains an executable closed loop policy, (2) it does not require any in-domain task-dependent supervision like expert labeled trajectories to either train or finetune the video trajectory generation or for policy training. Recent work [49] proposed an approach to generate the video plans without requiring in-domain supervision. But this work still requires training the RL agent for each given task prompt which can be tedious and time consuming.

**Zero-shot RL**: Zero-shot RL [77] promises the ability to quickly produce optimal policies for any given task defined by a reward function. A wide variety of methods have been developed to achieve zero-shot RL, which are in some ways generalizations of multi-task RL [12]. Most of these works assume a class of tasks where they can produce policies zero-shot. These tasks can be goal-conditioned [34, 19, 2, 65, 47], a linear span of certain state-features [15, 5] or some combination of some skills [20, 21, 56]. Recent works [84, 76, 77, 55, 1] employ a successor measure-based representation learning objective to be able to provide near-optimal policies for arbitrary reward function subject to model capacity constraints. Our work leverages these methods and finds the *best* reward supported by the representations that will produce the language-conditioned imagined trajectory.

# 3 Preliminaries

RLZero uses generative models to imagine trajectories from language prompts and directly produces a policy that results in observations imitating the imagined trajectory. In this section, we introduce the notion of trajectory generation, imitation learning, and unsupervised RL.

**Multimodal Video-Foundation Models (ViFMs) and In-Domain Video Generation:** Multimodal ViFMs [78, 79, 75] enable the understanding of video data in a shared representation space of other modalities such as text or audio. These shared representations can be used to condition video generation on different input modalities [36, 7]. Notably, these models can utilize text prompts to guide content, style, and motion, or employ an image as the initial frame for a subsequent video sequence. For this work, we use video generation models $VM$ that generate a sequence of video frames $\{i_1, i_2, ...i_n\}$ given a task specified in natural language $l$ by first converting the language prompt to a common embedding space across modalities; formally, $VM : l \rightarrow \{i_1, i_2, ...i_n\}$.

**Imitation Learning through Distribution Matching:** We consider a learning agent in a Markov Decision Process (MDP) [58, 71] which is defined as a tuple: $\mathcal{M} = (\mathcal{S}, \mathcal{A}, p, r, \gamma, d_0)$ where $\mathcal{S}$ and $\mathcal{A}$ denote the state and action spaces respectively, $p$ denotes the transition function with $p(s'|s, a)$ indicating the probability of transitioning from $s$ to $s'$ taking action $a$; $r$ denotes the reward function, $\gamma \in (0, 1)$ specifies the discount factor and $d_0$ denotes the initial state distribution. The reinforcement learning objective is to obtain a policy $\pi : \mathcal{S} \rightarrow \Delta(\mathcal{A})$ that maximizes expected return: $\mathbb{E}_\pi[\sum_{t=0}^\infty \gamma^t r(s_t)]$, where we use $\mathbb{E}_\pi$ to denote the expectation under the distribution induced by $a_t \sim \pi(\cdot|s_t), s_{t+1} \sim p(\cdot|s_t, a_t)$ and $\Delta(\mathcal{A})$ denotes a probability simplex supported over $\mathcal{A}$.

An imitation learning agent does not have access to the reward function, $R$, but has access to an "expert" trajectory (or a set of "expert" trajectories) from a policy that maximizes the reward function. Distribution matching objectives [25, 52] for imitation learning have been commonly used in some recent work [24, 66]. The objective in these methods is $\min_\pi \mathcal{D}(\rho^\pi, \rho^E)$, where $\rho^\pi$ is the visitation distribution of the policy $\pi$ (defined by the probability of being in state $s$ starting from the initial state distribution $s_0$ and following the policy $\pi$), $\rho^E$ is the visitation distribution exhibited by the "expert" trajectory and $\mathcal{D}$ is a function to compare the closeness of the distributions. $f$-Divergences are commonly used as a measure of distance between distributions.

**Successor Measure based Unsupervised RL:** Successor Measure [8] based learning has been recently studied [76, 1] as an unsupervised RL objective for its ability to describe long-term behavior

of the policy in the environment. Mathematically, successor measures define the measure over future states visited as $M^\pi$,

$$M^\pi(s, a, X) = \mathbb{E}_\pi \left[ \sum_{t \geq 0} \gamma^t p^\pi(s_{t+1} \in X | s, a) \right] \quad \forall X \subset \mathcal{S}. \tag{1}$$

Representing the successor measure for any policy $\pi$ as $\psi^\pi(s, a)^T \varphi(s^+)$ and trained on a dataset $d^O$ of offline interactions of the form $\{s, a, s'\}$, these methods facilitate extraction of a state-representation $\varphi(s)$ that is suitable for RL. Then, policies $\pi_z$(where the policies are represented using latents $z$) are learned offline that are near-optimal for a reward function defined in the span of learned state-features $r_z(s) = \varphi(s) \cdot z$. At test time, the policy for any given reward function $(r(s))$ can then be obtained with such a BFM (zero-shot, with no additional experiential data or gradient updates) using a closed-form solution to the following linear regression:

$$z_R = \min_z \mathbb{E}_{d^O} [(r(s) - \varphi(s) \cdot z)]^2 \implies z_R = \mathbb{E}_{d^O} [\varphi(s) \cdot \varphi(s)^T]^{-1} \mathbb{E}_{d^O} [\varphi(s) \cdot r(s)] \tag{2}$$

For this work, we use $(\varphi(s), \pi_z)$ to represent a BFM trained with successor-measure based unsupervised RL.

## 4 RLZero: Zero-Shot Prompt to Policy

RLZero uses components trained without any explicit in-domain supervision to map language to behaviors. For each domain, we consider a dataset $d^O$ of reward-free interactions $\{s, a, s'\}$ and a BFM $(\varphi(s), \pi_z)$ pre-trained on $d^O$. In the following sections, we describe the steps involved in detail. First, we present how an imagined trajectory is generated from a prompt. Then, we discuss how this imagined trajectory is projected to real observations of an agent. Finally, we describe the zero-shot procedure for inferring a policy that matches the behavior in the imagined trajectory. The latter two steps can also be used to directly infer policy when instead of language prompt, a cross-embodied video demonstration is provided.

### 4.1 Imagine: Unsupervised Generative Text-Conditioned Video Modeling

Grounding language to tasks has historically [26, 33, 54] required costly annotation labels that map language to task examples specified through image or state trajectories. Large video-language foundation models (ViFMs) help lift that requirement by training on vast amounts of internet videos, thus giving us a rich prior of grounding language commands to videos. We leverage the unsupervised video modeling approach from GenRL [49]. In the absence of any in-domain text labels, the approach uses an off-the-shelf video-language encoder with video encoder $f_v$ and language encoder $f_l$ trained on internet scale data to encode a sequence of observation frames $(o_{1:T})$ to a embedding space shared with its language description. A generative video model conditions on this latent embedding to generate a sequence of image observations $(i_{1:T})$. The video model $(VM)$ is trained to reconstruct the original observation frames $(o_{1:T})$ and at test time can simply be queried with a language instruction. Similar to GenRL, we use InternVideo2 as the video-language encoder and use a simplified GRU architecture for video generation. We refer the readers to GenRL for more details on training the video generation model and accounting for multimodality gap between video-text encodings. Training the video generation model does not require text labels mapping language to tasks and is fully unsupervised. Thus during inference, given a language instruction $e^l$, we obtain a sequence of frames $VM(f_l(e^l)) = (i_1, i_2...i_T)$ that represents an imagination of what the task looks like in the environment domain. Figure 6 shows an example of what these imaginings look like using the trained video generation model. Given the generated video of text description, GenRL learns an action-conditional world model to train a new policy for each text-instruction with model-based RL. This makes policy inference for a variety of tasks costly, unsafe, and memory-inefficient. In contrast, with RLZero we present a way to lift this limitation with unsupervised RL to directly infer a policy given generated video in the sections below.

### 4.2 Project: Grounding to Agent's Observation Space

The imaginings produced by $VM$ can be noisy, unrealizable, and not exactly representative of the domain. We propose to use semantic-similarity based retrieval to obtain nearest

frames in the dataset of the agent's prior environmental interactions $d^O$ to project *individual* frames of imagined trajectories to real observations as shown in Fig 2.

Semantic matching also allows us the flexibility to replace imaginings with a video demonstration of a task by a different embodiment agent in a different domain (e.g. as we will demonstrate in Section 5.2). In this work, we use an off-the-shelf vision-language embedding approach for retrieval, SigLIP [89], to map both the imagined frame and agent observation to the same latent embedding space, which has been pre-trained for similarity matching on an internet-scale dataset. As a single image observation is not informative of key state variables such as velocity or acceleration, we use the frame stacking

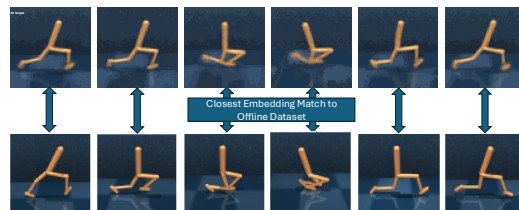

Figure 2: **Grounding Imagination in Real Observations**: We use nearest image retrieval defined by cosine similarity in the embedding space to output a real observation from the dataset that matches the imagined frame.

technique common in RL [50] when finding the nearest observation from the dataset corresponding to any imagined frame. We use an encoding function $\mathcal{E} : \mathcal{I} \to \mathcal{Z}$ to individually map a sequence of images to shared image-text embedding space.:

$$o_t = \arg\max_{o_t} \frac{\mathcal{E}(o_{t-k:t}) \cdot \mathcal{E}(i_{t-k:t})}{\|\mathcal{E}(o_{t-k:t})\|\|\mathcal{E}(i_{t-k:t})\|} \quad \forall t \in [T]. \tag{3}$$

Above we use $k$ previous frames (usually set to 3) to identify state variables in order to improve matching accuracy. Using the offline interaction dataset, we find corresponding proprioceptive states in addition to the real observation that we will subsequently use as a target sequence to imitate. We rely on proprioceptive states for the subsequent unsupervised RL phase, but in general BFMs may be trained with image observations if the state information is unavailable. While this step allows us to ground imagination to a sequence of real observations that the video expects the agent to follow, action inference still remains a challenge as the sequence might not be dynamically feasible or be available as a trajectory in the dataset to behavior clone.

### 4.3 Imitate: Zero-Shot Distribution Matching with Unsupervised RL

The resulting projected sequence of states might be dynamically infeasible and out of distribution from the trajectories in the offline dataset. Our key insight is to leverage unsupervised RL techniques to discover new skills from an offline dataset and use state-marginal matching to find a policy that induces the closest dynamically feasible sequence imitating the imagination. This section demonstrates that after pretraining a successor-measure based unsupervised RL agent with reward-free transitions, an approximate closed-form solution to state marginal matching exists without requiring any further gradient steps or interaction with the environment.

We take the distribution matching perspective of imitation learning [28, 25] and find a policy that matches the state marginal distributions of the grounded imagined trajectories (expert). Given a dataset of offline environmental interactions ($d^O$ with distribution $\rho$), we pretrain a successor-measures based unsupervised RL agent [1, 77] ($\varphi(s), \pi_z$) that provides us with a mapping from reward function $R$ to the corresponding $z_R$ that induces a near-optimal policy $\pi_{z_R}$ as shown in Eq. 2. For the case of state-marginal matching, finding the optimal policy reduces to finding the optimal latent $z$ minimizing the following distribution matching objective:

$$z_{imit} = \arg\min_{z} \mathcal{D}(\rho^{\pi_z}(s), \rho^E(s)), \tag{4}$$

where $\rho^E$ and $\rho^{\pi_z}$ are state visitation distribution of the "expert" imagined trajectories and of the policy $\pi^z$ respectively and $\mathcal{D}$ can be chosen to be mean-squared error, $f$-divergence, Integral Probability Metrics (IPM), etc. In general, minimizing the distance via gradient descent can provide a solution $z_{imit}$ to distribution matching. For the special case of KL divergence, Theorem 1 shows that $z_{imit}$ can be obtained using a learned distribution ratio between expert and offline interaction dataset $\rho^E/\rho$.

**Theorem 1.** *Define $J(\pi, r)$ to be the expected return of a policy $\pi$ under reward $r$. For an offline dataset $d^O$ with density $\rho$, a learned log distribution ratio: $\nu(s) = \log(\frac{\rho^E(s)}{\rho(s)})$, $D_{KL}(\rho^\pi, \rho^E) \leq$*

| | | Image-language reward (VLM-RM) | | Video-language reward | | GenRL | RLZero |
|---|---|---|---|---|---|---|---|
| | | IQL | TD3 (Base Model) | TD3 | IQL | model-based | |
| **Walker** | Lying Down | 2/5 (307.04±52.60) | — (116.97±69.70) | 2/5 (120.48±50.47) | **5/5 (419.05±281.56)** | 4/5 (199.66±30.64) | **5/5 (524.34±31.78)** |
| | Walk like a human | 1/5 (95.05±78.08) | — (30.34±34.58) | 3/5 (49.49±45.73) | 4/5 (146.86±54.40) | **5/5 (632.09±36.31)** | **5/5 (704.68±47.16)** |
| | Run like a human | **5/5 (50.93±5.05)** | — (61.76±52.49) | 1/5 (100.24±89.76) | 2/5 (286.14±63.91) | **5/5 (316.21±82.61)** | **5/5 (475.49±52.37)** |
| | Do lunges | 4/5 (130.98±109.87) | — (67.24±24.31) | 2/5 (48.71±12.06) | 3/5 (217.78±147.57) | **5/5 (484.68±50.60)** | **5/5 (377.53±30.14)** |
| | Cartwheel | 4/5 **(320.03±228.81)** | — (167.04±48.12) | 3/5 (151.16±67.95) | 4/5 (175.45±50.87) | **5/5 (252.68±33.76)** | 4/5 (254.89±12.74) |
| | Strut like a horse | **5/5 (110.08±50.24)** | — (164.61±110.45) | 1/5 (103.45±69.48) | 3/5 (81.14±25.01) | **5/5 (361.45±103.07)** | **5/5 (445.66±131.36)** |
| | Crawl like a worm | **4/5 (116.48±72.99)** | — **(94.54±16.18)** | 1/5 (84.43±36.97) | 2/5 (127.14±10.11) | 0/5 (127.48±24.81) | 3/5 (143.86±23.09) |
| **Quadruped** | Cartwheel | 1/5 (23.82±8.38) | — (6.81±3.24) | 3/5 (9.45±7.43) | 1/5 (14.07±5.33) | **4/5** (15.6±4.07) | **4/5 (30.88±0.95)** |
| | Dance | **5/5** (13.89±6.19) | — (8.19±4.74) | 3/5 (6.24±2.34) | 1/5 (11.79±4.28) | 4/5 (15.11±7.71) | **5/5 (24.29±3.29)** |
| | Walk using three legs | 2/5 (15.98±9.61) | — (6.70±3.61) | 2/5 (7.96±4.97) | 3/5 (7.46±1.75) | 4/5 **(24.70±6.56)** | **5/5 (27.29±7.29)** |
| | Balancing on two legs | 2/5 (13.92±5.52) | — (6.11±3.95) | 2/5 (6.69±8.19) | 2/5 (5.61±3.05) | **5/5** (7.20±3.78) | **5/5 (22.50±2.15)** |
| | Lie still | 1/5 **(82.93±77.13)** | — (9.51±7.54) | 3/5 (6.88±8.43) | 2/5 (67.96±34.91) | 3/5 **(135.26±57.41)** | 2/5 **(100.22±52.45)** |
| | Handstand | 2/5 (16.13±11.66) | — (4.18±0.88) | **4/5** (4.08±2.74) | 2/5 (6.80±2.85) | **4/5** (15.47±11.76) | 3/5 **(50.65±2.92)** |
| **Cheetah** | Lie down | **3/5** (219.35±58.77) | — (203.99±16.39) | 2/5 (255.03±49.28) | **3/5** (360.76±43.80) | **3/5 (468.52±17.45)** | 2/5 (202.67±34.75) |
| | Bunny hop | 3/5 (218.44±34.56) | — (181.74±35.52) | 1/5 (192.78±53.53) | 3/5 (129.26±1.37) | 4/5 (220.62±36.99) | **5/5 (224.70±6.46)** |
| | Jump high | 3/5 (172.10±39.02) | — (184.41±49.01) | 3/5 (183.36±73.40) | **5/5** (148.82±27.88) | **5/5 (267.54±28.17)** | **5/5 (232.23±41.65)** |
| | Jump on back legs and backflip | 3/5 (175.41±38.86) | — (197.62±42.52) | 0/5 (169.67±50.78) | 2/5 (131.14±15.24) | **5/5 (293.28±71.71)** | **5/5 (326.20±57.45)** |
| | Quadruped walk | 3/5 **(379.34±21.07)** | — (193.64±20.30) | 3/5 (187.24±46.67) | 3/5 (188.58±20.95) | 2/5 (318.10±37.53) | 4/5 **(388.02±37.76)** |
| | Stand in place like a dog | **4/5 (478.85±43.69)** | — (282.46±94.73) | 3/5 (238.58±88.26) | 0/5 (169.39±14.04) | 2/5 (238.25±47.83) | 3/5 (469.05±31.62) |
| **Stickman** | Lie down stable | 2/5 (201.03±82.97) | — (13.21±9.27) | 4/5 (30.86±4.60) | 1/5 (28.89±3.49) | **5/5 (686.56±386.66)** | 4/5 (841.48±226.24) |
| | Lunges | 0/5 (63.46±36.27) | — **(249.73±13.81)** | 2/5 (48.69±25.50) | 0/5 (38.77±3.69) | 4/5 **(244.82±58.80)** | **5/5** (191.41±61.51) |
| | Praying | 1/5 (50.84±22.27) | — (39.39±42.79) | 0/5 (49.13±32.51) | 0/5 (40.76±9.17) | 3/5 **(192.75±42.20)** | **4/5** (147.74±54.07) |
| | Headstand | 2/5 (20.14±12.24) | — (14.54±7.13) | 2/5 (48.11±42.51) | 1/5 (13.84±6.02) | **4/5** (71.75±32.77) | **4/5 (71.87±6.28)** |
| | Punch | 2/5 (73.42±19.00) | — (50.23±38.08) | 3/5 (74.73±7.01) | 4/5 (85.76±22.96) | **5/5** (181.08±69.58) | 4/5 (216.44±37.81) |
| | Plank | 0/5 (374.70±361.27) | — (507.46±289.41) | 0/5 (16.19±5.70) | 0/5 (66.62±53.43) | 1/5 (391.04±41.15) | 3/5 **(883.60±58.00)** |
| | Average | 51.2% (148.97) | Base Model (114.50) | 40% (87.75) | 44.8% (118.79) | 76.8% (246.48) | **83.2 % (295.11)** |

Table 1: Winrates computed by GPT-4o of policies output by different methods when compared to a base policies trained by TD3+Image-language reward. Bolded distribution-matching returns denote statistically significant improvement over the second best method under a Mann-Whitney U test with a significance level of 0.05.

$$-J(\pi, r^{imit}) + D_{KL}(\rho^\pi(s,a), \rho(s,a)) \text{ where } r^{imit}(s) = \nu(s) \ \forall s. \ \textit{The corresponding } z_{imit}$$
*minimizing the upper bound is given by* $z_{imit} = \mathbb{E}_\rho[r^{imit}(s)\varphi(s)] = \mathbb{E}_{\rho^E}[\frac{\nu(s)}{e^{\nu(s)}}\varphi(s)]$ *where* $\varphi$ *denotes state features learned by the BFM.*

Thus, with the reward function $r^{imit}(s) = \nu(s)$, we can use the solution of $z_{imit} = \mathbb{E}_\rho[\varphi(s)r^{imit}(s)]$ to retrieve the policy that mimics the grounded imagined behavior. However this reward function still requires learning a discriminator to obtain the distribution ratio for each task, which can lead to instabilities. A heuristic yet performant alternative is to use a shaped reward function $r(s) = e^{\nu(s)}$, similar to Pirotta et al. [57], which allows closed-form inference ($z_{imit} = \mathbb{E}_\rho[e^{\nu(s)}\varphi(s)] = \mathbb{E}_\rho[\frac{\rho^E(s)}{\rho(s)}\varphi(s)] = \mathbb{E}_{\rho^E}[\varphi(s)]$) without learning a discriminator, any gradient steps or interaction with the environment. We compare both approaches in Appendix C.5. The performance for both these methods are almost identical and we defer to the latter one (gradient-free inference approach) in all our experiments. The complete algorithm for RLZero can be found in Algorithm 1.

## 5 Experiments

Our experiments seek to understand the quality of behaviors that RLZero is able to produce in two settings (a) Language to Behavior, and (b) Cross-embodied Video to Behavior Generation. We note that none of the experiments below assume in-domain supervision such as annotations of trajectories in the environment with their task label, or expert demonstrations corresponding to the specified tasks either for training the video generation model or training the behavioral foundation model.

**Setup:** We consider continuous control tasks from the DeepMind control suite (Cheetah, Walker, Quadruped, Stickman) and HumEnv (3D Humanoid). The selected environments are diverse and complex, with their proprioceptive state space ranging from 17-358 dimensions and action space ranging from 6-69 dimensions. All of the environments use an environment horizon of 1000 steps, except HumEnv, which has a horizon of 300 steps. We use Cheetah, Walker, Quadruped, and Stickman environments to demonstrate language to behavior generation capabilities of RLZero and use Stickman and HumEnv environments to demonstrate cross-embodied abilities of RLZero to directly produce behaviors from out-of-distribution videos, such as those available on YouTube or AI-generated. Our choice of Stickman and HumEnv environments for cross-embodied behavior generation is motivated by the ease of availability of third-person individual human videos on internet as these environments reflect human morphology. We use an off-the-shelf pretrained BFM [74] trained for HumEnv with large MoCap datasets of real human behavior without assuming access to dataset used for training the BFM. This allows us to demonstrate the generality of the RLZero framework to plug a policy model trained with a different unsupervised RL algorithm to produce behaviors from cross-embodied videos. Our evaluation spans 25 tasks for language-to-behavior and 17 tasks for video-to-behavior.

**Implementation**: For the environments - Cheetah, Walker, Quadruped, and Stickman, we collect a dataset $d^O$ of environmental interactions of the form $\{s, a, s'\}$ purely by a random exploration algorithm RND [10] (10 million transitions for Stickman and 5 million transitions for the rest of the environments). In the Stickman environment, we additionally augment this dataset with replay buffers of the agent trained for walking and running to increase the diversity of behaviors, increasing the dataset size by another 2 million transitions. The detailed composition of the datasets can be found in Appendix B.3. We use the method from Section 4.1 to learn a text-conditioned video generation model using dataset $d^O$ *without requiring any in-domain task-trajectory labeling*. We use a successor-feature based unsupervised RL algorithm, Forward-Backward [76], to pretrain a general purpose policy BFM that can output a near-optimal policy given any reward function without test-time training or learning. For the projection step involved in language to behavior generation, we use SigLIP [89], an off-the-shelf image-text embedding model.

**Baselines:** For our evaluations, we consider the setting where the agent is not allowed environment interactions for learning during test time for a fair comparison and all baselines have access to same offline dataset $d^O$. This setting reflects the ability of the agents to use prior interaction data to learn meaningful behaviors. We compare against approaches that use foundation models trained with internet scale data to label rewards for offline transitions conditioned on text. These rewards can be computed using cosine similarity with text prompt per image observation of the agent as in VLM-RL [62] or per sequence of image observations using a video-embedding model. We use SigLIP as our image-text embedding model and InternVideo2 as a video-text embedding model for a fair comparison. Video-language embeddings take into account context and can potentially lead to more accurate reward estimation. Once the rewards are obtained, we use TD3 [23] and IQL [37] as the representative offline RL algorithms to obtain policies. We also compare to a recent approach GenRL, that performs model-based RL in the environment to learn

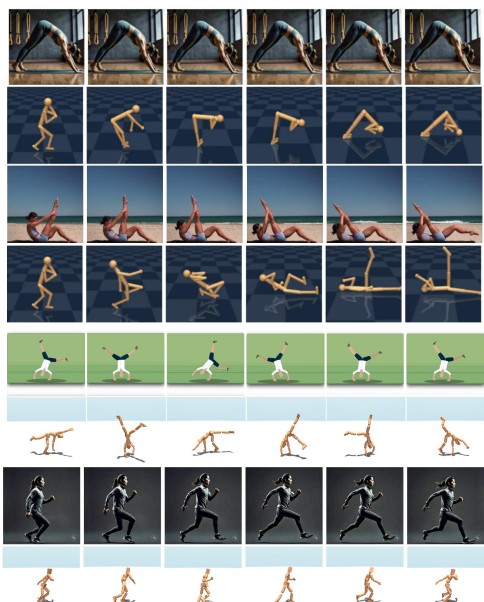

Figure 3: Examples for cross embodied imitation: `RLZero` can mimic motions demonstrated in YouTube or AI generated videos zero-shot. Top 2 rows: Stickman (2D Humanoid), Bottom 2 rows: SMPL 3D Humanoid.

a policy at test time by maximizing similarity to text-conditioned generated video. All baselines train a seperate policy for each task whereas `RLZero` trains a single policy for all tasks per environment using Forward-Backward unsupervised RL algorithm that uses TD3 as the underlying RL method.

**Evaluation Protocol**: We provide both qualitative and quantitative comparisons with baselines. The evaluation of behaviors generated given a text prompt or a video can be challenging as, unlike the traditional RL setting, we do not have access to a ground truth reward function. Instead, we have prompts that can be inherently ambiguous but reflect the reality of human-robot interaction. An obvious evaluation metric is to ask humans how much the generated behavior resembles their expectation of the behavior given the prompt. We use multimodal LLMs to qualitatively evaluate such preferences as a proxy to human preferences, as recent studies [14] have shown them to be correlated (up to 79.3%). To quantitatively compare the generated behaviors, we compute returns under a discriminator as a reward function that is trained to distinguish proprioceptive states corresponding to the projected frames of video (imagined or real) vs. offline dataset $d^O$ similar to [46].

## 5.1 Benchmarking Language to Continuous Control

The ability to specify prompts and generate agent behavior allows us to explore complex behaviors that might have required complicated reward function design. We curate a set of 25 tasks across 4 DM-control environments. Each of the agents has unique capabilities as a result of its embodiment,

and the prompts are specified to be reasonable tasks to expect for the specific domain. For each prompt, we generate behaviors for 5 seeds. The qualitative performance of any given method is evaluated as the win rate over the base method. We chose the base model for our comparisons as the policies trained via TD3 on image-language rewards. For each seed, we present the observation frames that the output policy generates and pass it to a Multimodal LLM capable of video understanding, which is used as a judge. Since the number of tokens can get quite large with the long default horizon of the agent (1000 frames), we subsample the videos by choosing every 8 frames and selecting the first 64 frames of size $256 \times 256$. We observed this subsampling to retain temporal consistency and the effective horizon ($8 \times 32 = 256$) to be long enough to demonstrate the task requested by the prompt. We consider a more fine-grained quantitative metric for comparison – average return under the distribution-matching reward function. We learn a discriminator between the projected states for a given imagination ($\rho^E$) and the offline interaction dataset $\rho$. Under the shaped reward function (Section 4.3), $r(s) = e^\nu(s) = \rho^E/\rho$ we compute returns for all the methods; a higher return indicates that a method is better able to match the projected imaginations.

Table 1 demonstrates the win rates by different methods when evaluated by GPT-4o-preview as well as distribution matching returns. We find that RLZero achieves a win rate of $83.2\%$ when compared to the best baseline (GenRL), which achieves a win rate of $76.8\%$. RLZero shows statistically significant improvement over baselines under distribution matching

| | Video Descriptions | SMODICE | RLZero |
|---|---|---|---|
| **Stickman** (2D Humanoid) | Human in backflip position | — $(16.05 \pm 3.44)$ | 2/5 $(14.27 \pm 0.02)$ |
| | Downward-facing dog yoga pose | — $(15.63 \pm 2.74)$ | 1/5 $(14.11 \pm 0.04)$ |
| | Cow yoga pose | — $(6.72 \pm 7.81)$ | **5/5 $(14.99 \pm 0.02)$** |
| | Downward dog one leg raised | — $(7.04 \pm 3.32)$ | **5/5 $(15.35 \pm 0.03)$** |
| | Lying on back, one leg raised | — $(9.07 \pm 2.07)$ | **5/5 $(12.55 \pm 0.01)$** |
| | Lying on back, both legs raised | — $(8.45 \pm 3.71)$ | **5/5 $(9.55 \pm 0.06)$** |
| | High plank yoga pose | — $(12.83 \pm 7.76)$ | **5/5 $(15.08 \pm 0.03)$** |
| | Sitting down, legs out front | — $(12.01 \pm 1.06)$ | **5/5 $(15.24 \pm 0.02)$** |
| | Warrior III pose | — $(17.27 \pm 5.09)$ | 3/5 $(15.22 \pm 0.01)$ |
| | Front splits | — $(13.44 \pm 2.39)$ | 4/5 $(\mathbf{15.21 \pm 0.01})$ |
| **HumEnv** (3D Humanoid) | A karate kick position | — $(50.02 \pm 0.02)$ | **5/5 $(199.90 \pm 0.01)$** |
| | A cat doing a handstand | — $(0.19 \pm 0.24)$ | **5/5 $(199.86 \pm 0.02)$** |
| | An arabesque ballet position | — $(10.02 \pm 0.01)$ | **5/5 $(199.92 \pm 0.04)$** |
| | Animated wikiHow cartwheel | — $(0.19 \pm 0.24)$ | **5/5 $(199.91 \pm 0.04)$** |
| | Running | — $(58.08 \pm 0.46)$ | **5/5 $(199.87 \pm 0.01)$** |
| | Lying crunches | — $(0.10 \pm 0.20)$ | **5/5 $(199.89 \pm 0.04)$** |
| | Plank position | — $(0.048 \pm 0.03)$ | **5/5 $(199.91 \pm 0.04)$** |

Table 2: Cross-Embodied Eval.: Distribution Matching Return & Winrates

returns. This indicates that RLZero is able to produce qualitatively better behaviors as well as do quantitatively better imitation than baselines. An important point to note is that all methods except RLZero requires test-time learning, whereas RLZero provides a gradient-free approach to instantly generating behaviors from text prompts. As an example, the closest performing baseline, GenRL requires test-time training with every task averaging $\approx$ 3-5 hours of training on NVIDIA-A40 GPU compared to our method, which requires $\approx$ 25 seconds to output a policy. The main bottleneck in RLZero is the projection step, which requires finding the closest real observation to each frame under a CLIP-like similarity metric and can be made significantly faster by using vector-databases specialized for retrieval. Figure 7 shows examples of behaviors output by RLZero on some of the prompts from our evaluation set. Additionally, all baselines require training a new policy model at test time for each prompt, as opposed to RLZero, which uses a single policy, making our approach parameter efficient.

## 5.2 Video to Continuous Control: Can RLZero succeed at cross-embodied imitation?

The intermediate stage in RLZero of matching the closest observations in the offline dataset to a frame from a video is based on semantic similarity. This means that RLZero can generalize to out-of-domain demonstrations building on the zero-shot generalization of CLIP-like representations. Subsequently, we can skip the *imagine* step completely if we are given an expert video demonstration. To investigate this, we consider a collection of videos scraped from Youtube as well as videos generated by open-source video generation tools like MetaAI and empirically test if RLZero is able to replicate the behaviors. We focus on Humanoid environments (Stickman, i.e 2D Humanoid, and HumEnv, i.e 3D Humanoid) here. For HumEnv [42], we use an open-source BFM [74] for policy generation given a reward function and do not assume access to the dataset used for training the BFM.

Table 2 shows the results for cross-embodied imitation across 17 video-clips. We use similar metrics to Section 5.1, but modified the GPT-4o prompt to take in the frames from the original video instead of a specified task description. We compare against SMODICE [46] which allows for using state-only observational data in conjunction with suboptimal offline data for imitation learning. This allows us to ablate the quality of imitation produced by a successor measure-based method that uses one policy model for all tasks as opposed to SMODICE which trains a new policy for each task. GenRL

requires a world model of the environment which is not available for HumEnv as we only have access to the pretrained policy. RLZero achieves a win rate of 80% against SMODICE for Stickman and 100% for HumEnv. This matches the observation from [57] that DICE-based methods lag behind in performance on observation-only imitation tasks. Figure 3 shows a qualitative comparison of the video and the obtained behavior on a few videos. Details about videos used can be found in Appendix C.3.

### 5.3 Ablation and Limitations

**Imagination-free behavior generation:** While the imagine, project, and imitate framework allows for the interpretability of the agent's behavior, we investigate if we can amortize the imagination and embedding search cost by directly mapping the language embedding to the skill embedding in the BFM's latent space. For this, we consider sampling $z$ uniformly in the latent space of the BFM and embedding the generated image observation sequence through a video-language encoder, which we denote by $e$. Given the observation sequence, we generate the $z_{imit}$ using the zero-shot inference process and learn a mapping from $e \rightarrow z_{imit}$ using a small 3-layer MLP. On the same tasks considered in Fig 3, we observe imagination-free RLZero to have a win rate of 65.71% over TD3 base model on Walker environment when compared to RLZero that had a win rate of $91.4\%$ (detailed

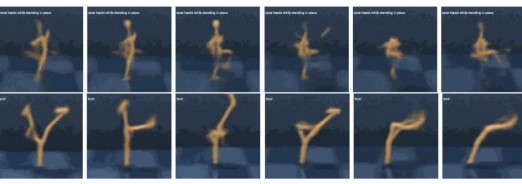

(a) Imagined behaviors: Top: stickman: 'raise hand while standing in place', Bottom: walker: 'kick'

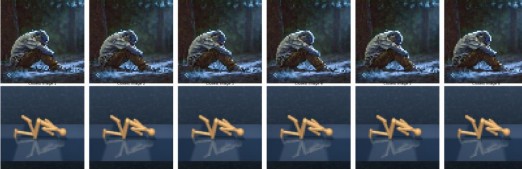

(b) Failed projection for a cross-embodied video.
Figure 4: Failure Cases in RLZero

results in Table 7). A thorough explanation of imagination-free RLZero can be found in Appendix B.6.

**Failures:** Our proposed method RLZero is not without failures. The stages of imagination and projection can fail individually, but the failures remain interpretable, i.e., by investigating the videos and the closest state match, we can comment on the agent's ability to faithfully complete that task to a certain extent.

**1. What I cannot imagine, I cannot imitate:** The video generation model used in our work is fairly small and limited in capability. We encountered limitations when generating complex behaviors with this model and found it to be sensitive to prompt engineering. Fortunately, as models get bigger and are trained on a larger set of data, this limitation can be overcome. Figure 4a shows some examples of these failures with the corresponding prompts.

**2. Limitation of semantic search-based image retrieval:** In this work, we used SigLIP, which has shown commendable performance for image retrieval tasks. We observed failure cases in the following scenarios (e.g. Figure 4b): a) Background distractors: We observe the image-similarity to latch on to features from the background and produce incorrect retrieval; b) Rough symmetries: In tasks where the agent is roughly symmetric (e.g. Walker when the head and legs are almost identical with a slight difference in width) the image retrieval fails by giving an incorrect permutation w.r.t the rough symmetries.

## 6 Conclusion

Specifying desired behavior to a learned agent is a longstanding problem in sequential decision-making. In this work, we take a step towards leveraging intuitive mediums such as language and cross-embodiment to specify desired policies. The RLZero framework bypasses costly annotations, reward hacking, and deployment time training by leveraging the rich capabilities of behavioral foundation models. Our introduced framework demonstrates a technique for directly translating from intuitive mediums for task specification to the latent space of policies without any test-time training. We demonstrate that RLZero not only recovers policies with zero gradient steps at evaluation time, but these policies also achieve qualitative and quantitative improvement over prior baselines that leverage VLMs as a source or reward or perform test-time training.

## Acknowledgments

We thank Matteo Pirotta, Ahmed Touati, Andrea Tirinzoni, Alessandro Lazaric and Yann Ollivier for enlightening discussion on unsupervised RL. This work has in part taken place in the Safe, Correct, and Aligned Learning and Robotics Lab (SCALAR) at The University of Massachusetts Amherst and Machine Intelligence through Decision-making and Interaction (MIDI) Lab at The University of Texas at Austin. SCALAR research is supported in part by the NSF (IIS-2323384), the Center for AI Safety (CAIS), and the Long-Term Future Fund. HS, SA, SP, MR, and AZ are supported by NSF 2340651, NSF 2402650, DARPA HR00112490431, and ARO W911NF-24-1-0193. This work has taken place in the Learning Agents Research Group (LARG) at the Artificial Intelligence Laboratory, The University of Texas at Austin. LARG research is supported in part by the National Science Foundation (FAIN-2019844, NRT-2125858), the Office of Naval Research (N00014-24-1-2550), Army Research Office (W911NF-17-2-0181, W911NF-23-2-0004, W911NF-25-1-0065), DARPA (Cooperative Agreement HR00112520004 on Ad Hoc Teamwork), Lockheed Martin, and Good Systems, a research grand challenge at the University of Texas at Austin. The views and conclusions contained in this document are those of the authors alone. Peter Stone serves as the Chief Scientist of Sony AI and receives financial compensation for that role. The terms of this arrangement have been reviewed and approved by the University of Texas at Austin in accordance with its policy on objectivity in research.

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

# Appendix

## A Proof For Theorem 1

**Theorem 1.** *Define $J(\pi, r)$ to be the expected return of a policy $\pi$ under reward $r$. For an offline dataset $d^O$ with density $\rho$, a learned log distribution ratio: $\nu(s) = \log(\frac{\rho^E(s)}{\rho(s)})$, $D_{KL}(\rho^\pi, \rho^E) \leq -J(\pi, r^{imit}) + D_{KL}(\rho^\pi(s, a), \rho(s, a))$ where $r^{imit}(s) = \nu(s) \ \forall s$. The corresponding $z_{imit}$ minimizing the upper bound is given by $z_{imit} = \mathbb{E}_\rho[r^{imit}(s)\varphi(s)] = \mathbb{E}_{\rho^E}[\frac{\nu(s)}{e^{\nu(s)}}\varphi(s)]$ where $\varphi$ denotes state features learned by the BFM.*

*Proof.* Let $\rho$ be the density of the offline dataset, $\rho^\pi$ be the visitation distribution w.r.t. policy $\pi$ and $\rho^E$ be the expert density. The distribution matching objective mentioned in Equation 4 using KL divergence is given as:

$$\min_{\rho^\pi} D_{KL}(\rho^\pi || \rho^E) \tag{5}$$

With simple algebraic manipulation, the divergence can be simplified to,

$$D_{KL}(\rho^\pi || \rho^E) = \mathbb{E}_{\rho^\pi}\left[\log \frac{\rho}{\rho^E}\right] + \mathbb{E}_{\rho^\pi}\left[\log \frac{\rho^\pi}{\rho}\right] \tag{6}$$

$$= \mathbb{E}_{\rho^\pi}\left[\log \frac{\rho(s)}{\rho^E(s)}\right] + D_{KL}(\rho^\pi(s) || \rho(s)) \tag{7}$$

$$= -J(\pi, \log \frac{\rho^E}{\rho}) + D_{KL}(\rho^\pi(s) || \rho(s)) \tag{8}$$

$$\leq -J(\pi, \log \frac{\rho^E}{\rho}) + D_{KL}(\rho^\pi(s, a) || \rho(s, a)) \tag{9}$$

The last line follows from the fact that $D_{KL}(\rho^\pi(s) || \rho(s)) \leq D_{KL}(\rho^\pi(s, a) || \rho(s, a))$.

$$D_{KL}(\rho^\pi(s, a) || \rho(s, a)) = \mathbb{E}_{\rho^\pi(s,a)}[\log \frac{\rho^\pi(s, a)}{\rho(s, a)}] \tag{10}$$

$$= \mathbb{E}_{\rho^\pi(s,a)}[\log \frac{\rho^\pi(s)\pi(a|s)}{\rho(s)\pi^D(a|s)}] \tag{11}$$

$$= \mathbb{E}_{\rho^\pi(s,a)}[\log \frac{\rho^\pi(s)}{\rho(s)}] + \mathbb{E}_{\rho^\pi(s,a)}[\log \frac{\pi(a|s)}{\pi^D(a|s)}] \tag{12}$$

$$= \mathbb{E}_{\rho^\pi(s)}[\log \frac{\rho^\pi(s)}{\rho(s)}] + \mathbb{E}_{\rho^\pi(s,a)}[\log \frac{\pi(a|s)}{\pi^D(a|s)}] \tag{13}$$

$$= D_{KL}(\rho^\pi(s) || \rho(s)) + \mathbb{E}_{s \sim \rho^\pi}[D_{KL}(\pi(a|s) || \pi^D(a|s))] \tag{14}$$

$$\geq D_{KL}(\rho^\pi(s) || \rho(s)) \tag{15}$$

Rewriting the minimization of the upper bound of KL as a maximization problem by reversing signs, we get:

$$\max_\pi \left[J(\pi, \log \frac{\rho^E}{\rho}) - D_{KL}(\rho^\pi(s, a) || \rho(s, a))\right] \tag{16}$$

The first term is an RL objective with a reward function given by $\log(\frac{\rho^E}{\rho})$, and the second term is an offline regularization to constrain the behaviors of offline datasets. Following prior works [35, 46], since our BFM is trained on an offline dataset and limited to output skills in support of dataset actions, and we can ignore the regularization to infer the latent $z$ parameterizing the skill. A heuristic yet performant alternative is to use a shaped reward function of $\frac{\rho^E}{\rho}$, which allows us to avoid training the discriminator completely and was shown to lead to performant imitation in [57]. $\qquad \square$

# B  Experimental Details

## B.1  Environments

### B.1.1  DM-control environments

We use continuous control environments from the DeepMind Control Suite [72].

**Walker:**  The agent has a 24 dimensional state space consisting of joint positions and velocities and 6 dimensional action space where each dimension of action lies in $[-1, 1]$. The system represents a planar walker.

**Cheetah:**  The agent has a 17 dimensional state space consisting of joint positions and velocities and 6 dimensional action space where each dimension of action lies in $[-1, 1]$. The system represents a planar biped "cheetah".

**Quadruped:**  The agent has a 78 dimensional state space consisting of joint positions and velocities and 12 dimensional action space where each dimension of action lies in $[-1, 1]$. The system represents a 3-dimensional ant with 4 legs.

**Stickman:**  Stickman was recently introduced as a task that bears resemblance to a humanoid in [49]. It has a 44 dimensional observation space and a 10 dimensional action space where each dimension of action lies in $[-1, 1]$.

**SMPL 3D Humanoid:**  The agent has a 358 dimensional state space consisting of joint positions and velocities and 69 dimensional action space where each dimension of action lies in $[-1, 1]$. The system represents a 3-dimensional humanoid.

For all the environments we consider image observations of size 64 x 64. All DM Control tasks have an episode length of 1000.

## B.2  Evaluation Protocol

To evaluate models for behavior generation through language prompts, we considered a set of 4 prompts per environment. One key consideration in designing these prompts was the generative video model's capability of generating reasonable imagined trajectories. Due to computing limitations, we were restricted to using a fairly small video embedding ( 1 billion parameters) and generation model ( 43 million parameters). The interpretability of our framework allows us to declare failures before they happen by looking at the generations for imagined trajectories.

For the set of task prompts specified by language, there is no ground truth reward function and there does not exist a reliable quantitative metric to verify which of the methods perform better. Instead, since humans communicate their intents via language, humans are the best judge of whether the agent has demonstrated the behavior they intended to convey. In this work we use a Multimodal LLM as a judge, following studies by prior works demonstrating the correlation of LLMs judgment to humans [14]. We use GPT-4o model as the judge, where the GPT-4o model is provided with two videos, one generated by a base method, and another generated by one of the methods we consider, and asked for preference between which video is better explained by the text prompt for the task. When inputting the videos to the judge, we randomize the order of the baseline and proposed methods to reduce the effect of anchoring bias. The prompt we use to compare the two methods is given here:

For prompt to policies:

```
response = client.chat.completions.create(
    model=MODEL,
    messages=[
    {"role": "system", "content": "For the given summarization:\
    '{task prompt}', which video is more aligned with the summarization?"},
    {"role": "user", "content": [
        "Video A",
        *map(lambda x: {"type": "image_url",
                        "image_url": {"url": f'data:image/jpg;base64,{x}'\
                        }}, video1),
        "Video B",
        *map(lambda x: {"type": "image_url",
```

```
13                        "image_url": {"url": f'data:image/jpg;base64,{x}'\
14                        }}, video2),
15         "FIRST provide a one-sentence comparison of the two videos\
16         and explain which you feel the given summarization explains better.\
17         SECOND, on a new line, state only 'A' or
18         'B' to indicate\
19         which video is better explained by the given \
20         summarization. Your response should use
21         the format:\
22         Comparison: <one-sentence comparison and explanation>\
23         Better explained by summarization: <'A' or 'B'>"
24         ]
25         }
26
27     ],
28 )
```

For cross-embodiment video to policies:

```
1  cross_embodied_video_description = [*map(lambda x: {"type": "image_url",
2      "image_url": {"url": f'data:image/jpg;base64,{x}'}},
3      cross_embodied_video)]
4
5  response = client.chat.completions.create(
6               model=MODEL,
7               messages=[
8               {"role": "system", "content": f"For the original video:
9               '{cross_embodied_video_description}', which of the
10              following given videos describe a behavior more similar
11              to the original video?"},
12              {"role": "user", "content": [
13                  "Video A",
14                  *map(lambda x: {"type": "image_url",
15                                  "image_url": {"URL":
16                      f'data:image/jpg;base64,{x}',
17                      }}, video1),
18                  "Video B",
19                  *map(lambda x: {"type": "image_url",
20                                  "image_url": {"URL":
21                      f'data:image/jpg;base64,{x}',
22                      }}, video2),
23                  "FIRST provide a one-sentence comparison of
24                  the two videos and explain \
25                  which you feel matches the behavior
26                  shown in original video better .
27                  SECOND, on a new line, state only 'A' or \
28                  'B' to indicate which video is better aligned
29                  to the task demonstrated in the original video.
30                  Your response should use \
31                  the format:\
32                  Comparison: <one-sentence comparison and explanation>\
33                  Better matches the original video: <'A' or 'B'>"
34                  ]
35                  }
36
37              ],
38          )
```

## B.3 Dataset Collection for Zero-Shot RL

For Cheetah, Walker, Quadruped, and Stickman environments, our data is collected following a pure exploration algorithm with no extrinsic rewards. In this work, we use intrinsic rewards obtained from Random Network Distillation [10] to collect our dataset based on the protocol by ExoRL [86] and using the implementation from repository ExoRL repository. For Cheetah, Walker,

and Quadruped, our dataset comprises 5000 episodes and equivalently 5 million transitions, and for Stickman, our dataset comprises 10000 episodes or equivalently 10 million transitions. Due to the high dimensionality of action space in Stickman, RND does not discover a lot of meaningful behaviors; hence we additionally augment the dataset with 1000 episodes from the replay buffer of training for a 'running' reward function and 1000 episodes of replay buffer trained on a 'standing' reward function.

## B.4 Baselines

Zero-shot text to policy behavior has not been widely explored in RL literature. However, Offline RL using language-based rewards utilizes an offline dataset to learn policies and is thus zero-shot in terms of rolling out the learned policy. This makes it a meaningful baseline to compare against. Offline RL uses the same MDP formulation as described in Section 3 to learn a policy $\pi : \mathcal{S} \to \Delta(\mathcal{A})$, given a reward function $r : \mathcal{S} \to \mathbb{R}$ and offline dataset $\mathcal{D}$. The offline dataset consists of state, action, next-state, reward transitions $(s, a, s', r(s))$. One of the core challenges of Offline RL is to learn a Q-function that does not overestimate the reward of unseen actions, which then at evaluation causes the agent to drift from the support of the offline dataset $\mathcal{D}$.

We implement two offline RL baselines to compare with RLZero– Implicit Q-learning (IQL, [37]) and Offline TD3 (TD3, [22]). Both of these methods share the same offline dataset as used to learn the successor measure in RLZero, which is described in Section 5, and gathered using RND. Since these datasets are reward-free, we must still construct a reward function that provides meaningful rewards for an agent achieving the behavior that aligns with the text prompt. Formally, given language instruction $e^l \in \mathcal{E}^l$, frame stack $(o_{t-k}, o_{t-k+1}, \ldots, o_t) \in \mathcal{I}$, and embedding VLM $\phi : \mathcal{E} \to \mathcal{Z}$, which can also embed frame stacks $\phi : \mathcal{I} \to \mathcal{Z}$ (and where observations $o_i \in \mathcal{I}$, and we use $o_i$ for this section), the reward for a corresponding language instruction and frame stack $k$ is the cosine similarity between the stacked language embedding and the frame embedding:

$$r(o_{t-k:t}, e^l) = \frac{\phi(e^l) \cdot \phi(o_{t-k:t})}{\|\phi(e^l)\| \|\phi(o_{t-k:t})\|} \tag{17}$$

For any individual task, $e^l$ is fixed and this is a reward function dependent on observations (as represented by a frame stack $o_{t-k:t}$). Notice that this representation closely matches that in Equation 3, but instead of finding the optimal sequence of observations, we simply compute reward as the cosine similarity between language and frames. Since the strength of the embedding space is vital to the quality of the reward function for offline RL, we evaluate two different vision-language models:

**Image-language reward** (SigLIP [89]): take a stack of 3 frames encode them using SigLIP, then the reward is computed as the cosine distance of the embeddings and the SigLIP embedding of language.

**Video-language reward** (InternVideo2 [79]): this method takes in previous frames $o_{0:t-1}$ as context and uses it to generate an embedding of the current frame observation $o_t$. The video encoder then takes the cosine similarity of $\phi(o_{0:t})$ and $\phi(e^l)$. This allows the reward function to provide rewards based not only on reaching certain states, but the agent exhibiting temporally extended behaviors that match the behavior. In practice, providing rewards using an image-based encoder for frame stacks can be challenging for tasks such as walking because they require context, and video-based rewards offer a way to better encode the temporal context.

### B.4.1 Offline RL

**Implicit Q-learning [37]** Implicit Q-learning builds on the classic TD error (revised in our context of language-instruction rewards):

$$L(\theta) = E_{(s,a,s',a') \sim \mathcal{D}}[(r(s, e^l) + \gamma Q_{\hat{\theta}}(s', a') - Q_\theta(s, a))^2]$$

to learn a Q function $Q_\theta$. IQL builds on this loss to handle the challenge of ensuring that the Q-values do not speculate on out-of-distribution actions while also ensuring that the policy is able to exceed the performance of the behavior policy. Exceeding the behavior policy is important because the dataset is collected using RND, meaning that any particular trajectory from the dataset is unlikely to perform well on a language reward. The balance of performance is achieved by optimizing the objective with expectile regression:

$$L_2^\tau(u) = |\tau - \mathbb{1}(u < 0)|u^2$$

Where $\tau > 0.5$ is the selected expectile. Expectile regression gives greater weight to the upper expectiles of a distribution, which means that the Q function will focus more on the upper values of the Q function.

Rather than optimize the objective with $Q(s', a')$ directly, IQL uses a value function to reduce variance to give the following objectives:

$$L_V(\psi) = E_{(s,a) \sim D}[L_2^\tau(Q_\theta(s, a)V_\psi(s))]$$

$$L_Q(\theta) = E_{(s,a,s',a') \sim \mathcal{D}}[L_2^\tau(r(s, e^l) + V_\psi(s') - Q_\theta(s, a))]$$

Using the Q-function, a policy can be extracted using advantage weighted regression:

$$L(\phi) = E_{(s,a) \sim \mathcal{D}}[\exp(\beta(Q_\theta(s, a) - V_\psi(s))) \log \pi_\phi(a|s)].$$

Where $\beta$ is the inverse temperature for the advantage term.

**TD3 [23]:**

TD3 was demonstrated to be the best performing algorithm when learning from exploratory RND datasets in [86]. While TD3 does not explicitly address the challenges discussed in implicit Q-learning and learns using Bellman Optimality backups, the approach is simple and works well in practice. The algorithm uses a deterministic policy extraction $\pi : \mathcal{S} \to \mathcal{A}$ to give the following objective:

$$\pi = \arg\max_\pi E_{(s,a) \sim \mathcal{D}}[Q(s, \pi(s))]$$

## B.5 RLZero

---
**Algorithm 1** `RLZero`

---
1: Init: Pretrained Video Generation Model $VM$, Pretrained BFM $\pi_z$, Offline Exploration Dataset $d^O$
2: Given: text prompt $t$
3: Generate imagination video given the text prompt: $\{i_1, i_2, ..i_l\} = VM(t)$
4: Project the imagined frames to real observations using embedding similarity as in Eq 3.
5: Use Theorem 1 for zero-shot inference to obtain BFM($\{s_1, s_2, ..., s_l\}$) = $z_{imit}$ and return $\pi_{z_{imit}}$.

---

### B.5.1 Text to Imagined Behavior with Video models

To generate a proposed video frame sequence, we utilize the GenRL architecture and provide the workflow using equations from the original paper [49]. First, the desired text prompt is embedded with the underlying video foundation model InternVideo2 [79] $e^{(l)} = f_{PT}^{(l)}(y)$. These embeddings are then repeated $n_{frames}$ times (we use $n_{frames} = 32$) to match the temporal structure expected by the world model. The repeated text embeddings are passed through an aligner module $e(v) = f_\psi(e^{(l)})$. The aligner is implemented as a UNet and it is used to address the multimodality gap [40] when embeddings from different modalities occupy distinct regions in the latent space. Next, the aligned video embeddings are concatenated with temporal embeddings. The temporal embeddings are one-hot encodings of the time step modulo $n_{frames}$ providing frame-level positional information. The first embedding is passed to the world model connector $p_\psi(s_t|e)$ to initialize the latent state. For each subsequent time step, the sequence model $h_t = f_\phi(s_{t-1}, a_{t-1}, h_{t-1})$ (implemented as a GRU) updates the deterministic state $h_t$. The deterministic state $h_t$ is mapped to a stochastic latent state $(s_t)$ using the dynamics predictor $p_\phi(s_t|h_t)$. The dynamics predictor, implemented as an ensemble of MLPs, predicts the sufficient statistics (mean and standard deviation) for a Normal distribution over $s_t$. During inference, the mean of this distribution is used as the latent state. Finally, the latent state $s_t$ is passed to a convolutional decoder $p_\phi(x_t|s_t)$ to reconstruct the video frame $x_t$. This process is repeated for all time steps ($t = 1, ..., n_{frames}$).

### B.5.2 Grounding imagined observations to observations in offline dataset

As described in Section 4, we ground imagined sequences by retrieving real offline states based on similarity in an embedding space. This enables us to create a suitable $z$-vector for distribution matching which is the expected value of the state features under the distribution of imagined states ($\rho_{imagined}$). During our dataset collection phase, we save both the agent's proprioceptive state as well as the corresponding rendered images and search over the images to then find the corresponding

state. Our code supports both stacked-frame embeddings and single-frame embeddings. We find that stacked-frame embeddings were helpful in modeling temporal dependencies through velocity and acceleration, which are crucial for recreating the intended behavior. SigLIP [89], which replaces CLIP's [60] softmax-based contrastive loss with a pairwise sigmoid loss, resulted in qualitatively better matches to exact positions within sequences, imitating behavior more accurately than CLIP. For both models, we use the OpenCLIP [30] framework. Our matching process first involves precomputing embeddings offline, which are stored in chunks of up to 100,000 frames to optimize memory usage and retrieval speed. During inference, we load this file and embed the query frame sequence from GenRL [49] into the same latent space. We process these query embeddings by dividing them into chunks of $k$-frame sequences ($k$ is generally 3 or 5), where each sequence consists of the current frame and the $k - 1$ preceding frames. If there are not enough preceding frames, we repeat the first frame to fill the gap. For each chunk of saved embeddings, we compute dot products between the query chunk and all subsequences of size $k$ in the saved embeddings. We track the highest similarity score for each query chunk and return the frames corresponding to the closest embedding sequences.

### B.5.3 Training a zero-shot RL agent

In this work, we chose Forward-Backward (FB) [76] as our zero-shot RL algorithm and trained it on proprioceptive inputs. Our implementation follows closely from the author's codebase . Specifically, FB trains Forward, Backward, and Actor networks. The backward networks are used to map a demonstration or a reward function to a skill, which is then used to learn a latent-conditional Actor. Our experiments were performed on NVIDIA-A40 and AMD EPYC 7763 64-Core Processor machine. The hyperparameters for our FB implementation are listed below:

**Implementation:** We build upon the codebase for FB `https://github.com/facebookresearch/controllable_agent` and implement all the algorithms under a uniform setup for network architectures and the same hyperparameters for shared modules across the algorithms. We keep the same method-agnostic hyperparameters and use the author-suggested method-specific hyperparameters. The hyperparameters for all methods can be found in Table 3:

Table 3: Hyperparameters for zero-shot RL with FB.

| Hyperparameter | Value |
|---|---|
| Replay buffer size | $5 \times 10^6$, $10 \times 10^6$ (for stickman) |
| Representation dimension | 128 |
| Batch size | 1024 |
| Discount factor $\gamma$ | 0.98 |
| Optimizer | Adam |
| Learning rate | $3 \times 10^{-4}$ |
| Momentum coefficient for target networks | 0.99 |
| Stddev $\sigma$ for policy smoothing | 0.2 |
| Truncation level for policy smoothing | 0.3 |
| Number of gradient steps | $2 \times 10^6$ |
| Regularization weight for orthonormality loss (ensures diversity) | 1 |
| **FB specific hyperparameters** | |
| Hidden units ($F$) | 1024 |
| Number of layers ($F$) | 3 |
| Hidden units ($b$) | 256 |
| Number of layers ($b$) | 2 |

### B.6 Imagination-Free `RLZero`

In this section, we propose an alternate method (Figure 5) for mapping a task description into a usable policy. Instead of first embedding a text prompt $e^\ell$, generating a video, then mapping the video to a policy parametrization, we propose to map the text prompt directly to a policy parametrization. To do this, we learn a latent mapper $m : e \rightarrow z_{\text{imitation}}$ that relates the latent space of a ViLM to the latent space of our policy parametrization. The mapper is a 3 layer MLP with hidden size of 512.

**Pretraining**: We first generate a dataset of episodes containing diverse behaviors by rolling out the

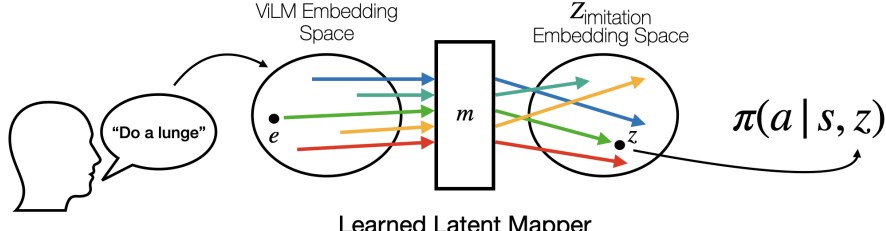

Figure 5: Illustrative diagram of imagination-free `RLZero` inference

behavior foundation model conditioned on a uniformly random sampled $z$. The resulting image observation sequences are then down-sampled (by 8) and sliced to break up each episode into smaller chunks of length 8; this preprocessing step helps increase the behavioral diversity and improves the ability of the ViFM to capture semantic meaning. The resulting clips are then embedded using a ViFM (InternVideo2 [79]) where each embedding is denoted by $e$ (as in Section 5.3). Now, we have a set of sequences of length 8 consisting of image observation along with their proprioceptive states, and the embedding for the image sequence.

Now, an obvious option is to map the embedding of image sequence to the $z$ that generated the trajectory. Unfortunately, the way BFMs are trained, they do not account for optimal policy invariance to reward functions. That is multiple reward functions that induce the same optimal policy are mapped to different encodings in the $\mathcal{Z}$-space. This presents a problem for the latent mapper, as it becomes a one-to-many mapping for any language encoding. We present an alternative solution which ensures that only one target $z$ is used for a given distribution of states induced by a language encoding. To achieve this we turn back to the imitation learning objective where the sequence of proprioceptive states is used to obtain a policy representation using Lemma 1 which gives the latent $z$ corresponding to the policy that minimizes the distribution divergence to the sequence of given states. We refer to the policy representation embedding space from the Forward-Backward representation as $\mathcal{Z}_{\text{imitation}}$-space.

When optimizing the latent mapper $m$, we minimize the following loss:

$$\mathcal{L}(\mathcal{D}, m) = \mathbb{E}_{(z_{\text{imitation}}, e) \sim \mathcal{D}} \left[ -\frac{m(e) \cdot z_{\text{imitation}}}{\|m(e)\| \cdot \|z_{\text{imitation}}\|} \right]$$

The latent space of the Backward representation is aligned with the latent space of the policy parametrization, so learning a mapping from the ViFM space to the Backward space is equivalent to learning a mapping from the ViFM space to the policy parametrization space.

**Inference:** During inference, the language prompt is embedded to a latent vector $e^l$. A known issue with multimodal embedding models is the embedding gap [40], which makes the video embeddings unaligned with text embeddings. To account for this gap, we use an aligner trained in an unsupervised fashion from previous work [49] to align the language embedding ($e^l_{aligned}$). Then the aligned embedding is passed through the latent mapper to get the policy conditioning $z_{imitation}$ which gives us the policy that achieves the desired behavior specified through language.

## C  Additional Results

### C.1  Visualizations

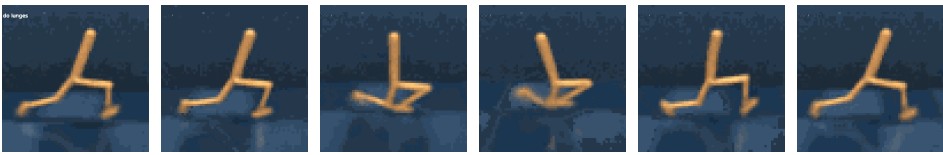

Figure 6: **Example Imagined Trajectories**: The video model imagines frames conditioned on the task specified as a text prompt 'do lunges'.

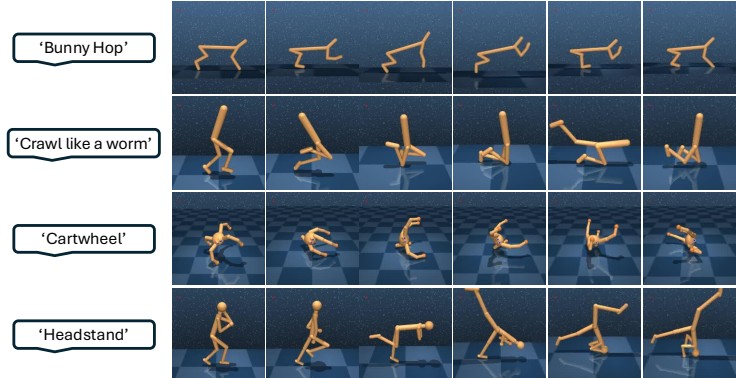

Figure 7: `RLZero` in action: Qualitative examples of RL converting the given language prompts into behaviors across different domains. Top to bottom: Cheetah, Walker, Quadruped, Stickman.

## C.2 Zero-shot imitation: Discriminator vs Discriminator-free

We experiment whether optimizing a tighter bound to KL divergence at the expense of training an additional discriminator in Lemma 1 leads to performance improvements. Table 4 shows that using a discriminator does not lead to a performance improvement and a training-free inference time solution achieves a slightly higher win rate.

## C.3 Cross Embodiment experiments

Tables 5 and 6 describe the videos used for cross-embodiment along with the win rate of the behaviors generated by `RLZero` when compared to a base model which trains SMODICE [46] on the nearest states found with the same grounding methods as `RLZero`.

## C.4 Imagination-Free RLZero Complete Results

We consider an ablation of our method by understanding the need for imagination by replacing the step with an end-to-end learning alternative. This is a novel baseline described in Appendix B.6. Table 7 shows the results of this end-to-end alternative which maps the shared latent space of video language models to behavior policy.

## C.5 Ablation on BFM

We perform an ablation on the choice of BFM. For RLZero, we used Forward-Backward Representations [76] to train the BFM but we experiment with an alternate method: Proto Successor Measures [1] for training BFMs. Table 8 shows that Forward Backward Representations performed better than Proto Successor Measures in our experiments.

## C.6 More Failure Cases

We include more failure cases in Figure 8 and Figure 9 as they can help in understanding the limitations of `RLZero` better and may inform future work.

## C.7 RLZero evaluation with Video-Embedding similarity

In this section, we experiment with another metric for comparison – embedding similarity between a video of the generated behavior and the text. We use InternVideo2 to embed the videos and take the cosine similarity with the prompt used to generate the behavior. Table 9 shows the results for this metric of comparison. Unfortunately, we observed that the similarity score is frequently higher even for behaviors that differ significantly from the prompt. This points to a limitation of using this metric for evaluation. Some reasons for this failure could be the limited context length of 8 for the video embedding model or a misalignment between video and text embedding vectors [40].

| | RLZero with discriminator | RLZero |
|---|---|---|
| **Walker** | | |
| Lying Down | 5/5 | 5/5 |
| Walk like a human | 5/5 | 5/5 |
| Run like a human | 5/5 | 5/5 |
| Do lunges | 5/5 | 5/5 |
| Cartwheel | 5/5 | 4/5 |
| Strut like a horse | 5/5 | 5/5 |
| Crawl like a worm | 1/5 | 3/5 |
| **Quadruped** | | |
| Cartwheel | 4/5 | 4/5 |
| Dance | 5/5 | 5/5 |
| Walk using three legs | 4/5 | 5/5 |
| Balancing on two legs | 4/5 | 5/5 |
| Lie still | 2/5 | 2/5 |
| Handstand | 4/5 | 3/5 |
| **Cheetah** | | |
| Lie down | 1/5 | 2/5 |
| Bunny hop | 5/5 | 5/5 |
| Jump high | 5/5 | 5/5 |
| Jump on back legs and backflip | 5/5 | 5/5 |
| Quadruped walk | 2/5 | 4/5 |
| Stand in place like a dog | 4/5 | 3/5 |
| **Stickman** | | |
| Lie down stable | 5/5 | 4/5 |
| Lunges | 5/5 | 5/5 |
| Praying | 4/5 | 4/5 |
| Headstand | 5/5 | 4/5 |
| Punch | 4/5 | 4/5 |
| Plank | 4/5 | 3/5 |
| Average | 82.4% | 83.2% |

Table 4: Win rates computed by GPT-4o of policies trained by different methods when compared to base policies trained by TD3+Image-language reward.

| | Prompt Descriptions | Video Link/Meta AI Prompt | Win rate vs SMODICE |
|---|---|---|---|
| **Stickman (2D Humanoid)** | Human in backflip position | animated human trying backflip | 2/5 |
| | Downward facing dog yoga pose | right profile of yoga pose downward facing dog | 1/5 |
| | Cow yoga pose | [51] | 5/5 |
| | Downward dog with one leg raised in the air | [51] | 5/5 |
| | Lying on back with one leg raised in the air | [53] | 5/5 |
| | Lying on back with both legs raised in the air | [41] | 5/5 |
| | High plank yoga pose | [81] | 5/5 |
| | Sitting down with legs laid in the front | [11] | 5/5 |
| | Warrior III pose | [87] | 3/5 |
| | Front splits | [70] | 4/5 |

Table 5: Comparison of Win rates vs SMODICE for Stickman

| | Prompt Descriptions | Video Link/Meta AI Prompt | Win rate vs SMODICE |
|---|---|---|---|
| **SMPL (3D Humanoid)** | A karate kick position | a karate kick | 5/5 |
| | A cat doing a handstand | a side profile of cat doing headstand | 5/5 |
| | An arabesque ballet position | ballet movement | 5/5 |
| | Animated wikiHow demo of a cartwheel | [83] | 5/5 |
| | Running | running | 5/5 |
| | Lying crunches | [82] | 5/5 |
| | Plank position | [81] | 5/5 |

Table 6: Comparison of Win rates vs SMODICE for 3D SMPL Humanoid

| Environment/Task | RLZero | RLZero (Imagination-Free) |
|---|---|---|
| **Walker** | | |
| Lying Down | 5/5 | 5/5 |
| Walk like a human | 5/5 | 4/5 |
| Run like a human | 5/5 | 1/5 |
| Do lunges | 5/5 | 5/5 |
| Cartwheel | 4/5 | 5/5 |
| Strut like a horse | 5/5 | 3/5 |
| Crawl like a worm | 3/5 | 0/5 |

Table 7: Win rates computed by GPT-4o of policies trained by different methods when compared to base policies trained by TD3+Image-language reward. `RLZero` shows marked improvement over using embedding cosine similarity as reward functions.

| | RLZero (PSM) | RLZero (FB) |
|---|---|---|
| **Walker** | | |
| Lying Down | 5/5 | 5/5 |
| Walk like a human | 5/5 | 5/5 |
| Run like a human | 2/5 | 5/5 |
| Do lunges | 4/5 | 5/5 |
| Cartwheel | 4/5 | 4/5 |
| Strut like a horse | 4/5 | 5/5 |
| Crawl like a worm | 1/5 | 3/5 |
| **Quadruped** | | |
| Cartwheel | 3/5 | 4/5 |
| Dance | 5/5 | 5/5 |
| Walk using three legs | 5/5 | 5/5 |
| Balancing on two legs | 4/5 | 5/5 |
| Lie still | 1/5 | 2/5 |
| Handstand | 4/5 | 3/5 |
| **Cheetah** | | |
| Lie down | 0/5 | 2/5 |
| Bunny hop | 5/5 | 5/5 |
| Jump high | 5/5 | 5/5 |
| Jump on back legs and backflip | 5/5 | 5/5 |
| Quadruped walk | 3/5 | 4/5 |
| Stand in place like a dog | 3/5 | 3/5 |
| Average | 71.6% | 84.2% |

Table 8: Win rates computed by GPT-4o of policies trained by different methods when compared to base policies trained by TD3+Image-language reward.

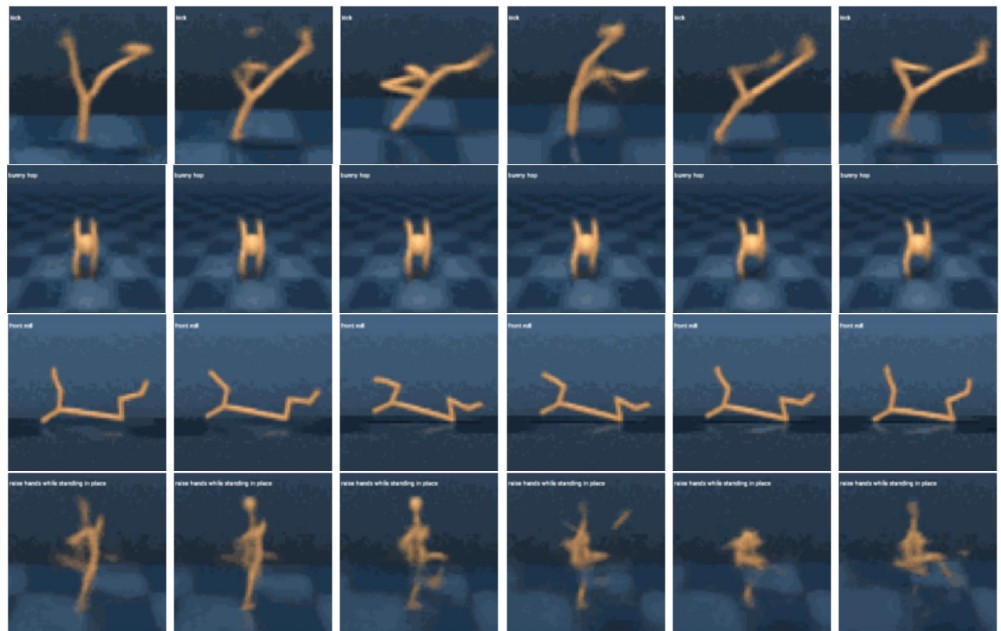

Figure 8: More examples of failed imagination by the video generation model used in `RLZero`. From top to bottom: Walker - 'kick', Quadruped - 'bunny hop', Cheetah - 'frontroll', Stickman - 'raise hands while standing in place'

| | Image-language reward | | Video-language reward | | RLZero |
|---|---|---|---|---|---|
| | IQL | TD3 (Base Model) | TD3 | IQL | |
| **Walker** | | | | | |
| Lying Down | 2/5 (0.95±0.00) | - (0.89±0.02) | 2/5 (0.93±0.01) | 5/5 (0.94±0.01) | 5/5 (0.93±0.00) |
| Walk like a human | 1/5 (0.93±0.00) | - (0.83±0.02) | 3/5 (0.92±0.01) | 4/5 (0.94±0.00) | 5/5 (0.98±0.00) |
| Run like a human | 5/5 (0.95±0.02) | - (0.88±0.03) | 1/5 (0.91±0.01) | 2/5 (0.94±0.00) | 5/5 (0.96±0.00) |
| Do lunges | 4/5 (0.94±0.01) | - (0.91±0.02) | 2/5 (0.92±0.00) | 3/5 (0.93±0.00) | 5/5 (0.94±0.01) |
| Cartwheel | 4/5 (0.95±0.01) | - (0.93±0.01) | 3/5 (0.94±0.01) | 4/5 (0.96±0.01) | 4/5 (0.95±0.01) |
| Strut like a horse | 5/5 (0.96±0.01) | - (0.94±0.02) | 1/5 (0.94±0.00) | 3/5 (0.96±0.03) | 5/5 (0.96±0.00) |
| Crawl like a worm | 4/5 (0.93±0.00) | - (0.92±0.01) | 1/5 (0.92±0.01) | 2/5 (0.95±0.01) | 3/5 (0.89±0.01) |
| **Quadruped** | | | | | |
| Cartwheel | 1/5 (0.95±0.00) | - (0.95±0.00) | 3/5 (0.95±0.01) | 1/5 (0.95±0.01) | 4/5 (0.92±0.02) |
| Dance | 5/5 (0.94±0.00) | - (0.94±0.00) | 3/5 (0.94±0.02) | 1/5 (0.94±0.01) | 5/5 (0.93±0.01) |
| Walk using three legs | 2/5 (0.92±0.00) | - (0.91±0.00) | 2/5 (0.91±0.01) | 3/5 (0.93±0.01) | 5/5 (0.93±0.01) |
| Balancing on two legs | 2/5 (0.93±0.01) | - (0.93±0.00) | 2/5 (0.93±0.01) | 2/5 (0.93±0.00) | 5/5 (0.94±0.02) |
| Lie still | 1/5 (0.87±0.01) | - (0.90±0.01) | 3/5 (0.94±0.00) | 2/5 (0.95±0.00) | 2/5 (0.92±0.00) |
| Handstand | 2/5 (0.91±0.01) | - (0.91±0.02) | 4/5 (0.92±0.01) | 2/5 (0.94±0.00) | 3/5 (0.91±0.00) |
| **Cheetah** | | | | | |
| Lie down | 3/5 (0.92±0.02) | - (0.87±0.00) | 2/5 (0.94±0.00) | 3/5 (0.94±0.01) | 2/5 (0.90±0.01) |
| Bunny hop | 3/5 (0.98±0.00) | - (0.98±0.00) | 1/5 (0.98±0.00) | 3/5 (0.97±0.02) | 5/5 (0.96±0.00) |
| Jump high | 3/5 (0.94±0.01) | - (0.94±0.01) | 0/5 (0.93±0.01) | 5/5 (0.93±0.01) | 5/5 (0.93±0.01) |
| Jump on back legs and backflip | 3/5 (0.93±0.01) | - (0.92±0.00) | 0/5 (0.91±0.01) | 2/5 (0.92±0.01) | 5/5 (0.91±0.01) |
| Quadruped walk | 3/5 (0.96±0.02) | - (0.85±0.01) | 3/5 (0.98±0.00) | 3/5 (0.99±0.01) | 4/5 (0.97±0.01) |
| Stand in place like a dog | 4/5 (0.93±0.01) | - (0.88±0.00) | 3/5 (0.98±0.01) | 0/5 (0.98±0.00) | 3/5 (0.97±0.00) |
| **Stickman** | | | | | |
| Lie down stable | 2/5 (0.92±0.00) | - (0.91±0.01) | 4/5 (0.93±0.00) | 1/5 (0.93±0.00) | 4/5 (0.91±0.00) |
| Lunges | 0/5 (0.92±0.00) | - (0.93±0.02) | 2/5 (0.92±0.01) | 0/5 (0.92±0.00) | 5/5 (0.96±0.00) |
| Praying | 1/5 (0.85±0.00) | - (0.89±0.02) | 0/5 (0.87±0.01) | 0/5 (0.87±0.01) | 4/5 (0.91±0.00) |
| Headstand | 2/5 (0.90±0.01) | - (0.90±0.00) | 2/5 (0.90±0.01) | 1/5 (0.87±0.01) | 4/5 (0.90±0.00) |
| Punch | 2/5 (0.89±0.02) | - (0.88±0.02) | 3/5 (0.88±0.02) | 4/5 (0.91±0.00) | 4/5 (0.90±0.02) |
| Plank | 0/5 (0.90±0.01) | - (0.93±0.03) | 0/5 (0.89±0.01) | 0/5 (0.93±0.00) | 3/5 (0.96±0.00) |
| Average | 51.2% (0.926) | Base Model (0.908) | 40% (0.927) | 44.8% (0.936) | 83.2% (0.933) |

Table 9: Win rates computed by GPT-4o of policies trained by different methods when compared to a base policies trained by TD3+Image-language reward. `RLZero` shows marked improvement over using embedding cosine similarity as reward functions.

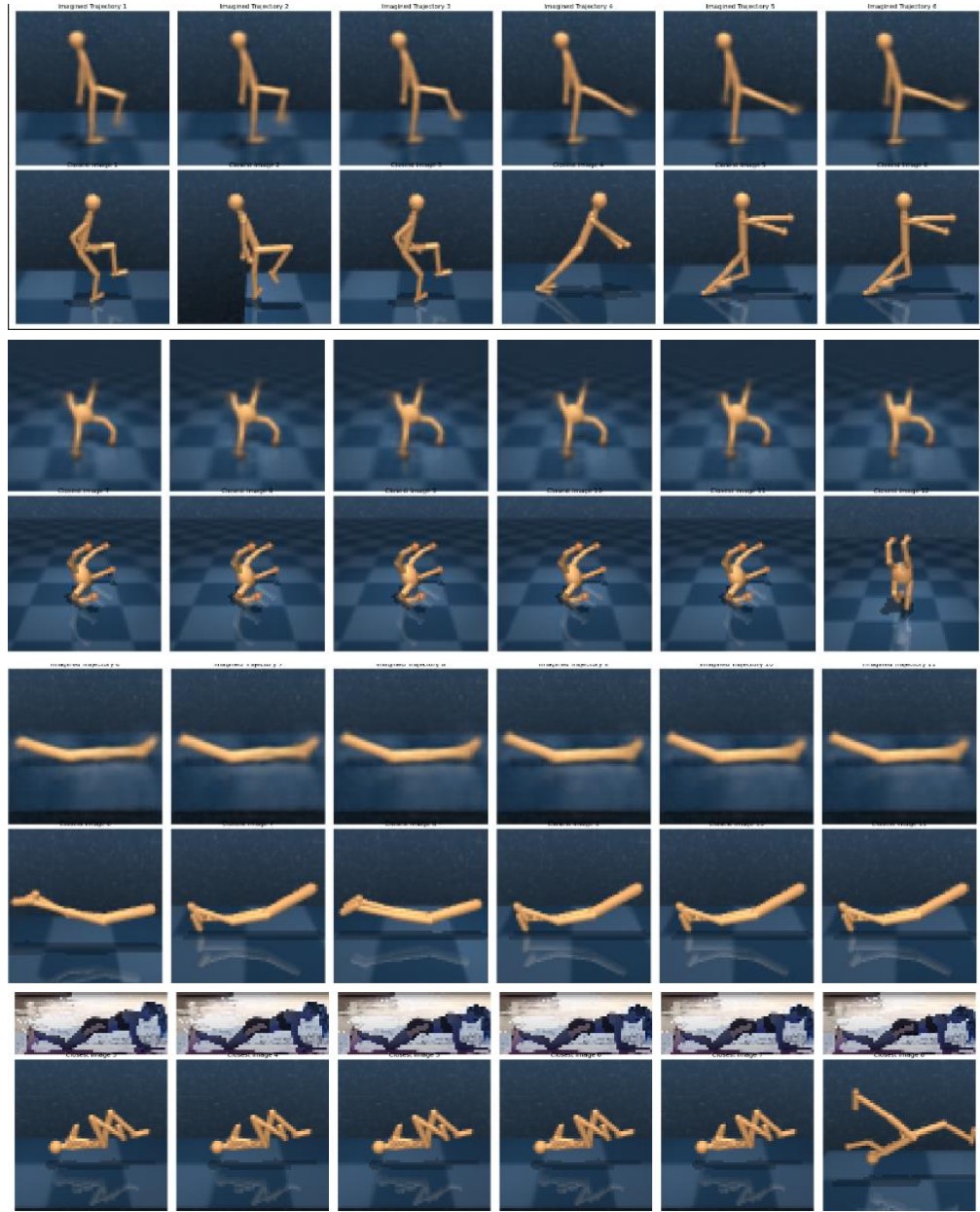

Figure 9: More examples of failed grounding by the image retrieval model used in RLZero. The top image shows the imagined frame or frame from the embodied video, and the bottom is the nearest frame obtained from the agent's prior interaction dataset.

