# OpenReview forum: "RLZero: Direct Policy Inference from Language Without In-Domain Supervision"
_NeurIPS.cc/2025/Conference — NeurIPS 2025 poster_

### Official Review · Reviewer_u2oR · 2025-06-15

**Clarity:** 2
**Significance:** 2
**Originality:** 2
**Rating:** 4
**Confidence:** 3

**Summary:**

The paper transfers a pre-trained reinforcement learning (RL) model to new tasks by aligning video of the target task with existing demonstration data.
It then uses a shaped reward function with a successor features model to update the imitation policy without gradient-based optimization.
The input video can come from a video-generation system or be collected from publicly available sources.
During alignment, the method selects the most similar short frame sequences between the target video and the demonstration dataset.
Experiments on continuous-control domains use videos generated from a text to video model or demonstrations from YouTube.
The approach outperforms GenRL (which relies on gradient updates) and avoids the need to train a separate policy for each new task, unlike the baseline RL methods evaluated.

**Questions:**

- For language-to-continuous control: why use win rate instead of score differences?
- How fast is the projection step in the various experiments?
	- How does it scale with more demonstrations or longer task videos?
	- How long were task videos?

**Ethical Concerns:**

["NO or VERY MINOR ethics concerns only"]

**Final Justification:**

The authors provided additional details on their evaluation methodologies and results. It is hard to tell whether the policies are succeeding in a ground truth sense, but the new results are modestly persuasive. I've revised my score somewhat accordingly.

**Limitations:**

Yes.

**Quality:**

3

**Strengths And Weaknesses:**

# quality
- Good to show reasonable baselines and report statistical differences.
	- Also to show why reward models are inefficient.
- Demonstrates reasonably good distribution matching
- Lacks metrics on task outcomes: the objectives are distribution matching rather than achieving an outcome. While this makes some sense for language commands, it would help if the commands aligned with task-based objectives in at least some cases.


# clarity
- Experiments would benefit from a clearer narrative and explicit summary of results / takeaways.


# significance
- Domain adaptation with new data is a useful technique.
- May be of interest in adapting models to new tasks through videos of demonstrations. Thus of potential interest to the robotics community.


# originality
- Mapping observations and mapping successor features are separately known, but the combination is novel. The novelty is in the steps of the process.

---

> ### Author Rebuttal · Authors · 2025-07-31
>
> We thank the reviewer for their thoughtful review. We answer the reviewers' questions below:
>
> > For language-to-continuous control: why use win rate instead of score differences?
>
> Our use of win-rate is motivated by the fact that there is no ground truth reward function associated with the prompted tasks. For instance, “walk like a human” may involve walking at 5 m/s or 10 m/s velocity. As a result, manually assigning a reward function can incorrectly favor one method over another. We use a denser metric in tables 1 and 2: discriminator as a reward function. This allows us to compare how well the agent is able to behave in a manner corresponding to the imagined trajectory for the language prompt and assign a dense score.
>
>
> > How fast is the projection step in the various experiments? a. How does it scale with more demonstrations or longer task videos? b. How long were task videos?
>
> The projection step takes a wall clock time of 10-20 seconds on AMD EPYC 7763 64-Core Processor CPU and 1 NVIDIA-A40 GPU. The costs involved here are forward inference to obtain the embedding of each frame in the video and then searching over the dataset, which requires O(n_video_frames x n_dataset_transitions) operations, thus scaling linearly with video length size. Note that the dataset embeddings are pre-computed and cached. The task videos were of variable lengths, from 36 frames to 120 frames.
>
>
>    > Lacks metrics on task outcomes: the objectives are distribution matching rather than achieving an outcome. While this makes some sense for language commands, it would help if the commands aligned with task-based objectives in at least some cases
>
>
>
> We appreciate the reviewers’ suggestion, but to reiterate from the first point above, that language to reward is inherently ambiguous, in particular for the domains we study in this work. As an example, language instructions of walking, strutting, or a yoga pose create ambiguity on what is the desired velocity, direction, height, etc. Looking at the output behaviors of one of the trained agents (RLZero, GenRL, VLM as reward) to get an outcome pose/goal pose would result in an unfair comparison.  We show one such comparison with hand-designed rewards below on some tasks - we compute outcome rewards for some tasks with hand-designed reward functions:
>
> | Task | TD3 (VLM-RM) | GenRL | RLZero |
> | --- | --- | --- | --- |
> | walker – lying down | 986.61 | 989.6 | 1000.0 |
> | walker – Walk like a human | 151.91 | 570.13 | 704.63 |
> | quadruped – handstand | 7.99 | 139.5 | 210.53 |
>
>
> We would also like to clarify that we do not use a demonstration dataset, as mentioned in the reviewer’s summary.  Our dataset is collected mostly by random exploration in language-to-policy experiments. This makes learning imitation policy challenging from offline data.
>
>
> Please let us know if our clarifications resolve your questions and concerns.

---

> > ### Comment · Reviewer_u2oR · 2025-08-05
> >
> > Thank you for the clarifications.
> >
> > > We appreciate the reviewers’ suggestion, but to reiterate from the first point above, that language to reward is inherently ambiguous, in particular for the domains we study in this work.
> >
> > My thinking was that there are tasks that have reward functions that can _also_ have a language description (perhaps generated by a VLM or similar model from the video). This would then offer a more direct test of whether the language instruction is enabling execution of a desired behavior.
> >
> > > We would also like to clarify that we do not use a demonstration dataset, as mentioned in the reviewer’s summary. Our dataset is collected mostly by random exploration in language-to-policy experiments. This makes learning imitation policy challenging from offline data.
> >
> > Apologies for the misunderstanding. I struggled to follow many parts of the text details, as noted by the other reviewers.
> >
> > > Note that the dataset embeddings are pre-computed and cached. The task videos were of variable lengths, from 36 frames to 120 frames.
> >
> > I did not realize how short these sequences might be. This is a good part to emphasize in the text.

---

> ### Author Response · Authors · 2025-08-06
>
> We thank the reviewer for following up on this. We answer the raised questions below:
> > My thinking was that there are tasks that have reward functions that can also have a language description (perhaps generated by a VLM or similar model from the video). This would then offer a more direct test of whether the language instruction is enabling execution of a desired behavior.
>
>
> Note that the tasks we provided in the comment above are actually the ones from our evaluation benchmark that we found to have the least ambiguous closed-form reward from DMC-control corresponding to the language prompt.
>
> We stress that a language to reward mapping is rarely one-to-one and so the methods we compare might generate a policy that optimizes one reward function but not another and choosing one of those reward functions might result in a cherry picked evaluation.
>
> The classes of tasks that we are looking at might have complex reward functions (such as a backflip). While the language models might be able to generate some reward functions for these tasks, they are not guaranteed to be robust. Moreover, the corresponding policies generated by optimizing the reward functions might hack the rewards and no longer describe the desired behavior. The challenge of aligning reward functions to human desired behaviors has been widely studied [1, 2] and this problem would be exacerbated by complex rewards and tasks.
>
> [1]: Booth, Serena, et al. "The perils of trial-and-error reward design: misdesign through overfitting and invalid task specifications." Proceedings of the AAAI Conference on Artificial Intelligence. Vol. 37. No. 5. 2023.
>
> [2]: Knox, W. Bradley, et al. "Reward (mis) design for autonomous driving (abstract reprint)." Proceedings of the AAAI Conference on Artificial Intelligence. Vol. 38. No. 20. 2024.
>
> > Apologies for the misunderstanding. I struggled to follow many parts of the text details, as noted by the other reviewers.
>
> Based on the other reviewer BfVk’s suggestion on giving high level overview of sections, we have updated the paper to give high level overview before we dive into details. We note our changes below:
> 1. Currently, the introduction describes how video models can be out-of-distribution of the agent’s observation space and thus we require projection to in-domain observations. Then we explain how to translate from projected frames to a policy. We understand that though this was how we went about this research, a reader might appreciate a high-level overview first. We have modified the second paragraph at line 41 to add “We achieve this through an imagine-project-imitate process, where ‘imagination’ transforms language to images, ‘projection’ from images to agent observations and ‘imitate’ from agent observations to policies,” and we can continue to fine-tune the writing.
> 2. In line 144-151 of the method section we explain a high-level overview of our approach, but our method subsections directly dive into low-level details.
> We have added the following three-step list to better clarify the method.
> * Imagination: utilize a video language model to perform the mapping from a descriptive sentence to a sequence of images: $VL(f_l(e^l))\rightarrow (i_1,i_2,...,i_T)$, where $f_l$ is a language embedder)
>
> * Projection: Transform the individual frames from video into a sequence of states grounded in the agent’s observation space using similarity: $(\{o_t = \text{argmax}_{o_t \in \mathcal O} SigLIP(o_t, i_t) \quad \quad \forall t \in \{1,..., T\}\})$.
>
> * Imitation: Use the zero-shot behavior model to produce the projected behavior by extracting the policy parameterized by $z$ that induces the state occupancy implied by $\rho^E \leftarrow (o_1,o_2,..., o_T)$.
>
>
>
> > I did not realize how short these sequences might be. This is a good part to emphasize in the text.
>
> A number of state-of-the-art closed-source models generate videos that are only 5-10s in length at the moment and we relied on an open-source video generation model in this work. These models are expected to improve substantially, and those improvements will directly improve RLZero. We clarify that our agent horizon is 1000 timesteps, so we are still outputting an imitation policy for the entire horizon. We will add this description in the text and thank the reviewer for pointing this out.

---

> > ### Comment · Reviewer_u2oR · 2025-08-07
> >
> > Thank you for the additional remarks.
> >
> >
> > > We show one such comparison with hand-designed rewards below on some tasks - we compute outcome rewards for some tasks with hand-designed reward functions:
> >
> > > Note that the tasks we provided in the comment above are actually the ones from our evaluation benchmark that we found to have the least ambiguous closed-form reward from DMC-control corresponding to the language prompt.
> >
> > I'm not sure how to understand the previous table. Were those tasks where the language reward was based on the text in the table? And what does "hand-designed rewards" mean here? The rewards from the environments as defined? Something written specifically for this?

---

> > > ### Author Response · Authors · 2025-08-07
> > >
> > > To clarify the results in the table, the following describes our process:
> > >
> > > 1. We used 3 total reward functions for which an optimal policy looks like "lying down:", "walk like a human", and "handstand." We used the standard reward for Walker task "walk" which produces policies that resemble the language task "walk like a human" and we hand-designed rewards for the "lying down" and "handstand" tasks specifically for this evaluation. **So as not to bias our results, we did not use RLZero behavior as a reference to design our reward functions, but rather we used our intuition for these environments (e.g., lying down requires the height of the chest to be small).**
> > > 2. We prompted RLZero and two baselines (GenRL, and TD3-RM) with the language commands.
> > > 3. We evaluated the resulting trajectories using the hand-designed reward.
> > >
> > > We believe this evaluation demonstrates that our method out-performs baselines on both standard and hand-designed rewards for tasks that also have a convenient language description.

---

### Official Review · Reviewer_f371 · 2025-06-25

**Clarity:** 2
**Significance:** 4
**Originality:** 3
**Rating:** 5
**Confidence:** 3

**Summary:**

This work proposes an imagine, project, and imitate framework to enable zero-shot language-to-behavior grounding. The transfer depends on a pretrained general video generation model, a projection function (semantic similarity), and an embodiment-specific unsupervised RL training stage. Crucially, none of the components need language-annotated in-domain demonstrations: the foundation models used for video generation and image similarity search are pretrained on general, scalable datasets, and the successor measure based unsupervised RL can be trained on mixed-quality offline interactions in the target domain. The offline interaction dataset is also used by the semantic retrieval step in order to ground the plan into the target observation space. Either a language command or a cross-embodiment video can be used to specify a task, with experimental evaluation in DeepMind control and HumEnv simulation environments.

**Questions:**

1. How exactly is the video generation model adapted to the offline dataset? Do you adopt the Aligner and Connector ideas from GenRL, even if you do not learn a world model? Moreover, why not condition the video generation to start from the current image observation from the environment?
2. Is the assumption of the reward being linear with respect to the pretrained $\varphi(s)$ a limitation?
3. Can your reward definition also be hacked? Such as by visiting the expert’s state distribution but doing so out-of-order?
4. Why do you refer to GenRL training as less safe than RLZero?
5. Would a vision-only embedding model work for the projection step or does the model need to be trained for multimodal matching?

**Ethical Concerns:**

["NO or VERY MINOR ethics concerns only"]

**Final Justification:**

While minor issues around clarity remain, they don’t affect my overall positive assessment. I maintain my acceptance rating.

**Limitations:**

yes

**Quality:**

3

**Strengths And Weaknesses:**

**Strengths**:
- A neat way to use video generative models as a prior for task specification, and especially powerful when combined with unsupervised RL.
- Presents several desirable qualities over prior vision-language reward models, such as supporting immediate policy extraction, requiring less in-domain supervision, and reducing the risk of reward hacking.
- Only (or mostly) unsupervised in-domain data is needed.
- A policy can be derived in closed form without gradient updates or additional interaction, leading to impressive policy generation speed.
- The proposed framework is interpretable: failures can be inspected for trajectory imagination and projection separately.

**Weaknesses**:
- Random exploration with RND seems to not quite be sufficient to produce relevant state space coverage in Stickman, which is why the offline data needed to be augmented with walking and running demonstrations (not annotated with natural language, however). The problem of generating an informative exploration dataset for complex domains remains a key issue.
- Excluding GenRL as a baseline in HumEnv doesn’t seem entirely justified simply because the authors choose to start from a pretrained BFM rather than the offline dataset that was used to create the BFM.
- The chosen discriminator-based quantitative evaluation metric may be biased in favour of the proposed method. Specifically, the discriminator is trained to consider projected frames as high-scoring behavior and the offline dataset as low-scoring. In this light, it might be expected that RLZero would perform well on this metric as it directly uses projected frames obtained in a similar way as an imitation target, whereas the baselines do not.
- The paper could be improved by making the explanations more self-contained. In particular, a corresponding appendix section or a specific reference would help clarify how the functions $\psi$ and $\varphi$ are trained (and how a policy is derived from them). Same with how the video generation model is adapted with the offline dataset.

**Minor**:

Small issues with the notation:
- d_0 and s_0 are both used to denote the initial state distribution
- it seems to be redundant to use both expectation and probability in Eq. 1
- s+ is not defined
- in Eq. 2, I believe the square should be inside the expectation

---

> ### Author Rebuttal · Authors · 2025-07-31
>
> We thank the reviewer for their detailed feedback. We respond to the reviewer’s concerns and questions below:
>
> > Random exploration with RND seems to not quite be sufficient to produce relevant state space coverage in Stickman, which is why the offline data needed to be augmented with walking and running demonstrations (not annotated with natural language, however). The problem of generating an informative exploration dataset for complex domains remains a key issue
>
> We appreciate the reviewer's observation and acknowledge in the paper that state-coverage remains a key challenge for unsupervised learning. For any unsupervised method, the skills learned are limited to data coverage, and investigating how to enable scalable data collection is an interesting research topic, but beyond the scope of the work.
>
>
> > Excluding GenRL as a baseline in HumEnv doesn’t seem entirely justified simply because the authors choose to start from a pretrained BFM rather than the offline dataset that was used to create the BFM.
>
> Following the reviewers’ suggestion, we aim to include the comparison to GenRL for the HumEnv environment by using the dataset provided by MetaMotivo. The HumEnv environment requires learning a world model for a complex 3D humanoid, and our initial experiment resulted in a collapsed world model. Training the world model is a computationally intensive task (5 days on single GPU [1]) and might require longer time than the rebuttal to ascertain the hyperparameters that work for this setting. Since our language-to-policy experiments utilize a common video model to generate imaginations, our comparisons already show that RLZero is competitive to GenRL while bypassing test-time training and being faster at inference.
>
> > The chosen discriminator-based quantitative evaluation metric may be biased in favour of the proposed method.
>
> The return under discriminator reward has been shown to be a valid metric to measure the distribution matching ability as it is an upper bound to KL divergence between the imitator and expert’s state visitation distribution[2,3] rather than a hand-crafted one. While methods based on image-language rewards or video-language rewards use a proxy objective for imitation, our method directly optimizes the objective we actually care about. As a result of this, our method and the GenRL baseline are expected to optimize the discriminator better. We also compare win-rates, which we believe is a truly unbiased metric to gain understanding of the effect of the entire pipeline from language-to-behaviors.
>
> > The paper could be improved by making the explanations more self-contained.
>
> We appreciate the reviewers’ suggestion and will update the appendix with more details on unsupervised RL training and video generation model training. We will also fix the notation issues that the reviewer pointed out. Note that NeurIPS does not allow updating the paper during the rebuttal period, but we will ensure to make these changes in the final version of the paper.
>
> > How exactly is the video generation model adapted to the offline dataset? Do you adopt the Aligner and Connector ideas from GenRL, even if you do not learn a world model? Moreover, why not condition the video generation to start from the current image observation from the environment?
>
> Yes, we follow the unsupervised video model training method from GenRL. Conditioning the video on the current image observation is an interesting idea that can enable closed-loop control with fewer compounding errors and is an interesting avenue for future research.
>
> > Is the assumption of the reward being linear with respect to the pretrained \phi(s) a limitation?
>
> No we do not believe reward being linear with respect to pretrained  \phi(s)  to be a limitation. Note that we are learning rewards that are linear in terms of arbitrarily non-linear features of states - this can be viewed as a neural network that is learned to predict rewards from state, and we take the last layer of the NN to be features of the state under which the reward function remains linear.
>
> > Can your reward definition also be hacked? Such as by visiting the expert’s state distribution but doing so out-of-order?
>
> Yes, we agree the way we define the reward function can potentially be hacked in the sense of permutation of states if allowed by the dynamics of environment. While we did not observe this to be a limitation in the environments we considered, this can be fixed in a principled way by matching visitations on {s,s’} and requires minimal changes to training the BFM.
>
> > Why do you refer to GenRL training as less safe than RLZero?
>
> RLZero allows us to partly interpret the output of the policy before deploying the policy by looking at the closest projected images. Tasks which do not seem realistic/unsafe can be discarded this way; in GenRL we know policy outcome only after deploying the learned policy.
>
> > Would a vision-only embedding model work for the projection step or does the model need to be trained for multimodal matching?
>
> Since we are interested only in image-to-image semantic similarity, it is possible that vision-only embedding would work, but language provides task information that might assist in better embedding similarity computation.
>
>
>
> [1]: Mazzaglia, Pietro, et al. "GenRL: Multimodal-foundation world models for generalization in embodied agents." Advances in neural information processing systems 37 (2024): 27529-27555.
>
> [2]: Ma, Yecheng, et al. "Versatile offline imitation from observations and examples via regularized state-occupancy matching." International Conference on Machine Learning. PMLR, 2022.
>
> [3]: Zhu, Zhuangdi, et al. "Off-policy imitation learning from observations." Advances in neural information processing systems 33 (2020): 12402-12413.

---

> > ### Comment · Reviewer_f371 · 2025-08-01
> > **Acknowledgement of rebuttal**
> >
> > Thank you for the answers. I have read them and will consider them when finalizing my review.
> >
> > I encourage the authors to complete the experiment on HumEnv as discussed.
> >
> > While the linearity of rewards based on pretrained phi(s) and the unsupervised video model training method from GenRL answers are not entirely clear, they don’t affect my overall positive assessment.

---

> ### Author Response · Authors · 2025-08-07
>
> We thank the reviewer for their response and appreciate the reviewer for the positive assessment. We acknowledge the suggestions provided by the reviewer to further improve our paper. We address some of the concerns below:
> 1. Learning a video model in GenRL for a 3D humanoid environment is challenging, and the video generations we attempted do not reflect the cross-embodied video demonstration given as input in the space of 3D humanoid observation. This inherently reflects the brittleness and the extra computation required of the GenRL pipeline but we are still attempting to train a stable video generation model in the setting of 3D humanoid using GenRL pipeline that we hope to have in the final version of the paper. We reiterate that as a result of using the same underlying video model in Section 5.1, Table1 compares RLZero and GenRL in a head-to-head setting for video-to-policy in addition to language-to-policy. Our results already demonstrate that RLZero is a competitive baseline for the video-to-policy setting while requiring no test-time training.
>
> 2. The unsupervised RL methods we use, which are based on successor features, assume a set of reward functions for which they can achieve zero-shot policy inference. This reward distribution is given by the set: $( r = \phi z | z \in \mathbb{S}^d )$, where $\phi$ are the state features and $z \in \mathbb{S}^d$ represents d-dimensional vectors $z$ being sampled from a d-dimensional norm ball. While these reward spaces might look restrictive, good quality feature sets (which is provided by Forward-Backward representations) span a wide and diverse set of reward functions that can describe a large number of the tasks. We also note that in our experiments dimensionality of $\phi(s)$ is set to 128 for all environments in Section 5.1 which is greater than the raw observation dimensionality of the corresponding environment.
>
> 3. We use the exact same training procedure for video generation as in GenRL without any modifications, but to make the paper more self-contained will add a short description of the video-generation training mechanism in the appendix (as it is not our contribution). The model takes in a language embedding transformed by aligner to reduce the multimodal gap between text and video embeddings, and use it to predict the video sequence. The aligner is trained by assuming that text and video embeddings are separated by small amount due to noise.

---

### Official Review · Reviewer_BfVk · 2025-07-01

**Clarity:** 2
**Significance:** 3
**Originality:** 3
**Rating:** 5
**Confidence:** 4

**Summary:**

The paper presents RLZero, which trains a video foundation model to generate videos given a reward-unlabeled offline dataset. This video model is used at test time to generate videos of an agent performing a task, given a language description, and then project the videos to the dataset to find nearest matching states. A state-distribution matching objective is applied to learn a policy to imitate these behaviors.

**Questions:**

When projecting generated observaiton sequences to closest matches in the offline dataset, won’t some of the closest-match observation sequences come from different trajectories? If so, there could be some discontinuities in the states retrieved, making the state distribution matching objective valid but potentially suboptimal?

**Ethical Concerns:**

["NO or VERY MINOR ethics concerns only"]

**Final Justification:**

The authors have addressed my concerns during the rebuttal. Results are good and motivation makes sense. I am recommending an accept.

**Limitations:**

Yes

**Quality:**

3

**Strengths And Weaknesses:**

## Strengths

**Analysis:** THere’s some nice analysis and discussion on inference time of the pipeline.

**Results:** Results across many tasks in a few embodiments demonstarte RLZero works pretty well. Analysis is comprehensive too, and the reported metrics make sense given the open-ended evaluation.

**Method:** I like that the method intelligently combines the best aspects of large foundation models (to use for training the video gen model and perform projection) while also combining with more traditional RL techniques such as successor feature representations and state distribution matching.

## Weaknesses

**Writing Clarity:**

- The intro doesn’t really flow well from one point to the next. It’s fine, but I think the logic takes a couple of turns by presenting a multi-step solution that spans paragraphs 2 and 3. Perhaps hinting at (1) imagining and (2) projecting far earlier; e.g., paragraph 1, would help with understanding the intro.
- Similarly the methods section is written at an extremely low level without having a high-level to refer to. It feels that each part of the method is written after the previous, but there’s no high-level intuition about how the method works at the top of each subsection (4.2 and 4.3). This could just be me, but the intro and methods took me multiple times to read to fully understand the contributions of the paper and how exactly it works.

**Lots of prior data:**

- Experiments are performed with 5-12million initial transitions. This is a LOT of data. Some experiments with less data would be beneficial to demosntrating the scalability o the method.

## Minor Things

Some related work in learning generalist agents that can immediately produce actions for a given language description:

- BOSS: https://clvrai.github.io/boss/
- https://arxiv.org/abs/2306.01711
- DECKARD: https://deckardagent.github.io/

---

> ### Author Rebuttal · Authors · 2025-07-31
>
> We thank the reviewer for their detailed feedback on our work. We address the reviewers' questions and concerns below:
>
> > The intro doesn’t really flow well from one point to the next. It’s fine, but I think the logic takes a couple of turns by presenting a multi-step solution that spans paragraphs 2 and 3. Perhaps hinting at (1) imagining and (2) projecting far earlier; e.g., paragraph 1, would help with understanding the intro.
>
> We will reword the introduction to give an overview of the imagine and project steps before diving into some of the details. Note that NeurIPS does not allow updating the paper during the rebuttal period, but we will ensure to make these changes in the final version of the paper.
>
>
> > Similarly the methods section is written at an extremely low level without having a high-level to refer to. It feels that each part of the method is written after the previous, but there’s no high-level intuition about how the method works at the top of each subsection
>
> We will also provide a high level overview of our method before diving into details in the method sections. We thank the reviewer for pointing this out.
>
>
> > Experiments are performed with 5-12million initial transitions. This is a LOT of data
>
>
> Some of the most sample-efficient model-based learning algorithms require 500k learning steps to learn a *single* task (Figure 3 in [1]). On the other hand, we are leveraging a model-free algorithm to learn a continuous space of tasks (effectively an infinite set of tasks) parameterized by a latent vector z in a single policy model. By learning a continuous space of tasks, in fact the model is more efficient in learning by leveraging a relatively moderate number of transitions to learn a large continuous space of behaviors.
>
>
>
> > When projecting generated observaiton sequences to closest matches in the offline dataset, won’t some of the closest-match observation sequences come from different trajectories? If so, there could be some discontinuities in the states retrieved, making the state distribution matching objective valid but potentially suboptimal?
>
> The reviewer brings up an interesting point about closest matches coming from different trajectories. We highlight two key properties of the method: 1. The objective of state marginal matching does not require realizability. Our analytical solution is an outcome of ‘best-effort’ distribution matching – by finding the observation trajectory that is dynamically feasible and minimizes the KL divergence to the projected trajectory 2. Unsupervised RL allows one to stitch together behaviors and learn behaviors going beyond the trajectories in the dataset. In particular, it optimizes behaviors that maximize a parameterized class of reward functions given an offline dataset, solutions of which may require stitching multiple sub-trajectories.
>
> > Related Works
>
> We thank the reviewer for bringing the works to our attention and we will add a discussion on them in our related work section.
>
> Please let us know if our clarifications resolve your questions and concerns.
>
> [1]: Hansen, Nicklas, Xiaolong Wang, and Hao Su. "Temporal difference learning for model predictive control. Proceedings of the 39 th International Conference on Machine Learning

---

> > ### Comment · Reviewer_BfVk · 2025-08-05
> >
> > Thanks for the response. I'll keep my borderline accept score for now since most of the issues were with writing quality and the authors promised to change things for the camera-ready (but did not point out specifics) so there's no way to double check that it will be the case.
> >
> > I am still overall positive about the paper and think it should get in.

---

> ### Author Response · Authors · 2025-08-06
>
> We thank the reviewer for the positive assessment. We explain in more detail how we rewrote the introduction section:
> 1.  Currently, the introduction describes how video models can be out-of-distribution of the agent’s observation space and thus we require projection to in-domain observations. Then we explain how to translate from projected frames to a policy. We understand that though this was how we went about this research, a reader might appreciate a high-level overview first. We have modified the second paragraph at line 41 to add “We achieve this through an imagine-project-imitate process, where ‘imagination’ transforms language to images, ‘projection’ from images to agent observations and ‘imitate’ from agent observations to policies.”  and we continue to fine-tune the writing.
> 2. In the method section, line 144-151 explain a high-level overview of our approach, but our method subsections directly dive into low-level details.
> We have added the following three-step list to better clarify the method.
>
> * Imagination: utilize a video language model to perform the mapping from a descriptive sentence to a sequence of images: $VL(f_l(e^l))\rightarrow (i_1,i_2,...,i_T)$, where $f_l$ is a language embedder)
>
> * Projection: Transform the individual frames from video into a sequence of states grounded in the agent’s observation space using similarity: $(\{o_t = \text{argmax}_{o_t \in \mathcal O} SigLIP(o_t, i_t) \quad \quad \forall t \in \{1,..., T\}\})$.
>
> * Imitation: Use the zero-shot behavior model to produce the projected behavior by extracting the policy parameterized by $z$ that induces the state occupancy implied by $\rho^E \leftarrow (o_1,o_2,..., o_T)$.

---

> > ### Comment · Reviewer_BfVk · 2025-08-08
> >
> > Thanks for the clarification on how the writing will be changed. I am satisfied with the changes and will keep these in mind during the post-rebuttal discussion.

---

### Official Review · Reviewer_uCNH · 2025-07-01

**Clarity:** 3
**Significance:** 2
**Originality:** 3
**Rating:** 4
**Confidence:** 4

**Summary:**

This paper proposes to leverage foundation models for policy learning by first imagining a trajectory conditioned on language instructions, then projecting this imagined trajectory to real observations from the dataset, and finally imitating those observations using distribution matching. They show results on language-conditioned continuous control tasks from DM Control Suite and on video-conditioned continuous control tasks on Humanoid environments (for cross-embodiment transfer).

**Questions:**

See Strengths and Weaknesses

**Ethical Concerns:**

["NO or VERY MINOR ethics concerns only"]

**Final Justification:**

The authors addressed my concerns and therefore I increased my score.

**Limitations:**

Yes

**Quality:**

3

**Strengths And Weaknesses:**

Strengths
- Leveraging off-the-shelf foundation models for continuous control is a promising approach to address the scarcity of data for control, and it is especially interesting that compared to other methods this does not require in-domain trajectories for learning inverse dynamics models
- The paper is clearly written
- The results on the humanoid tasks, which are more realistic than DM Control, show that the method can be quite general

Weaknesses
- The approach relies heavily on the “project” step which can be extremely brittle. The authors themselves acknowledge this: “We observe the image-similarity to latch on to features from the background and produce incorrect retrieval.” It would be informative to see results on more “in-the-wild” videos with noisier backgrounds.
- A strong alternative to semantic similarity would be using off-the-shelf pose detection models and matching the joints. Could the authors compare to those methods?

---

> ### Author Rebuttal · Authors · 2025-07-31
>
> We thank the reviewer for their detailed feedback. We appreciate the reviewer for acknowledging that our method is promising in sparse data regimes without requiring in-domain demonstrations, and finding our paper to be clearly written and our method to be quite general. We discuss some of the highlighted weaknesses below:
>
> > Project step can be brittle
>
> As the reviewer notes correctly, the projection step can be brittle as we are aiming to match semantic information about poses that are relevant for the embodiment. While we explicitly highlight this as a limitation in Section 5.3, our work opens up new avenues for better semantic matching relevant to robot control. An example improvement would be to use a multimodal language model to obtain conditional embeddings that can be prompted to match the relevant parts. Our use of SigLIP/CLIP style approach goes to show that even unconditional embeddings can lead to performant imitation when learning from relatively clean third-person videos.
>
>
>
> > A strong alternative to semantic similarity would be using off-the-shelf pose detection models and matching the joints
>
> While we agree that this could be an interesting experiment, our work and baselines share the common foundation of not assuming access to embodiment-specific models. Our unsupervised learning approach only assumes access to reward-free environment interactions of the specific agent and proposes an approach to enable zero-shot imitation with general foundation models. A number of hurdles make use of the proposed pose detection approach limiting: 1. Video and robot may have similar yet differing embodiments/skeletons, like different numbers of joints, 2. Pose detection models require domain-specific models and a domain-specific projection step involving complex retargeting [1,2] 3. Matching 3D skeleton in 2D images can create false positives exploiting symmetries.
>
> [1]: Liu, Junjia, et al. "Human-Humanoid Robots Cross-Embodiment Behavior-Skill Transfer Using Decomposed Adversarial Learning from Demonstration." arXiv preprint arXiv:2412.15166 (2024).
>
> [2]: Cao, Zhe, et al. "Openpose: Realtime multi-person 2d pose estimation using part affinity fields." IEEE transactions on pattern analysis and machine intelligence 43.1 (2019): 172-186.

---

> > ### Comment · Reviewer_uCNH · 2025-08-06
> >
> > Thanks for the response. I have increased my score to 4.

---

### Note · Authors · 2025-08-16

This work introduces low-level action inference from language with Behavior Foundation Models (BFM) without any additional training during test time. This is achieved by using unsupervised RL to map video to a latent task embedding. Our experiments compare on domains with up to 358 and 69 dim. state and action spaces, respectively.

In particular, as the reviewers point out, RLZero:
- (**uCNH, f371**) does not require in-domain expert trajectories (u2oR)
- (**uCNH**) scales to complex embodiments
- (**BfVk**) is evaluated on many tasks
- (**BfVk,f371**) combines traditional RL with foundation models
- (**f371**) [allows] policy failures [to] remain interpretable.

### Addressing Reviewer Concerns

1. Writing changes:

**Rev. BfVk, u2oR** suggest giving overviews before diving into low-level details.

We have added a high-level overview in the method section and introduction, which is detailed in **Rev. BfVk’s** rebuttal.

Additionally, we add more details on video models and BFMs from prior work in the suppl. to make the paper more self-contained.

2. **Rev. f371** recommended we run the GenRL baseline for cross-embodied video to policy in HumEnv

For HumEnv (3D env with large state-action space), we used an off-the-shelf BFM. GenRL requires training a video gen. model from offline transitions. We attempted to use publicly available HumEnv transition data to train a video model, but the model resulted in collapsed predictions, presumably due to the complexity of the environment. In table 1, we note that both GenRL and RLZero use the same video model to generate imagination based on language, so we already see that RLZero is competitive to GenRL when imitating from videos with the additional benefit of not requiring test-time training.

3. **Rev. u2oR** suggested we additionally compare on some outcome-based rewards

Our main evaluation criteria with which we compare methods is winrate as decided by GPT-4o, along with distribution matching performance to imagination, rather than a return from some “ground-truth” reward function. It is rare for an arbitrary language task to be described well with a concise reward function (eg, run like a human may describe running at many different velocities). However, at u2oR’s request, we conduct further evaluation of our policies on simple tasks with known/intuitive reward functions, and find our method still outperforms policies trained with our closest baselines.

We thank the reviewers and AC for their feedback.

---

### Decision · Program_Chairs · 2025-09-17

**Decision:**

Accept (poster)

**Comment:**

This paper introduces RLZero, a framework for zero-shot language-to-policy inference that does not require in-domain supervision or labeled trajectories. The method follows an imagine–project–imitate pipeline: a video model imagines a trajectory from language input, a projection step aligns it with the target environment’s observation space, and an unsupervised RL agent instantly imitates the projected trajectory through closed-form policy inference. The work is evaluated on DM Control Suite and humanoid domains, including cross-embodiment transfer from third-person videos.

The reviewers agree that the main strength lies in the novelty of combining foundation video models with unsupervised RL to bypass costly language-annotated or in-domain demonstrations. The paper demonstrates generality across tasks and embodiments, immediate policy extraction without gradient updates, and interpretability of failures through the modular pipeline. Results are competitive with strong baselines such as GenRL, and the approach shows promise for scaling to real-world settings where supervision is scarce.

The main weaknesses highlighted include (i) brittleness of the projection step, which sometimes latches onto irrelevant features, (ii) reliance on large offline datasets (millions of transitions), which raises questions about scalability, (iii) limited comparisons on outcome-based metrics, and (iv) clarity issues in the introduction and method presentation. Some reviewers also noted missing baselines (GenRL on HumEnv) and potential bias in the evaluation metric. The authors responded constructively, adding high-level overviews for clarity, clarifying evaluation methodology, and providing additional outcome-based results to address reviewer concerns. While some limitations remain, particularly around projection robustness and dataset efficiency, the consensus is that these are natural challenges for this early-stage approach rather than fatal flaws.

Overall, this is a technically solid paper with interesting and original contributions at the intersection of RL, generative video modeling, and language grounding. It does not yet rise to the level of a spotlight due to evaluation and clarity limitations, but it represents a meaningful step forward and will be of broad interest to the NeurIPS community.